# Self-directed online machine learning for topology optimization

Changyu Deng [1], Yizhou Wang [2], Can Qin[2], Yun Fu[2] & Wei Lu [1,3✉]

Topology optimization by optimally distributing materials in a given domain requires non-gradient optimizers to solve highly complicated problems. However, with hundreds of design variables or more involved, solving such problems would require millions of Finite Element Method (FEM) calculations whose computational cost is huge and impractical. Here we report Self-directed Online Learning Optimization (SOLO) which integrates Deep Neural Network (DNN) with FEM calculations. A DNN learns and substitutes the objective as a function of design variables. A small number of training data is generated dynamically based on the DNN's prediction of the optimum. The DNN adapts to the new training data and gives better prediction in the region of interest until convergence. The optimum predicted by the DNN is proved to converge to the true global optimum through iterations. Our algorithm was tested by four types of problems including compliance minimization, fluid-structure optimization, heat transfer enhancement and truss optimization. It reduced the computational time by 2 ~ 5 orders of magnitude compared with directly using heuristic methods, and outperformed all state-of-the-art algorithms tested in our experiments. This approach enables solving large multi-dimensional optimization problems.

[1] Department of Mechanical Engineering, University of Michigan, Ann Arbor, MI 48109, USA. [2] Department of Electrical and Computer Engineering, Northeastern University, Boston, MA 02115, USA. [3] Department of Materials Science and Engineering, University of Michigan, Ann Arbor, MI 48109, USA. ✉email: weilu@umich.edu

Distributing materials in a domain to optimize performance is a significant topic in many fields, such as solid mechanics, heat transfer, acoustics, fluid mechanics, materials design and various multiphysics disciplines[1]. Many numerical approaches[2] have been developed since 1988, where the problems are formulated by density, level set, phase field, topological derivative, or other methods[3]. Typically, these approaches use gradient-based optimizers, such as the Method of Moving Asymptotes (MMA), and thus have various restrictions on the properties of governing equations and optimization constraints to allow for fast computation of gradients. Because of the intrinsic limitation of gradient-based algorithms, the majority of existing approaches have only been applied to simple problems, since they would fail as soon as the problem becomes complicated such as involving varying signs on gradients or non-linear constraints[4]. To address these difficulties, non-gradient methods have been developed which play a significant role in overcoming the tendency to be trapped in a local minimum[5].

Non-gradient optimizers, also known as gradient-free or derivative-free methods, do not use the gradient or derivative of the objective function and have been attempted by several researchers, most of which are stochastic and heuristic methods. For instance, Hajela et al. applied Genetic Algorithm (GA) to a truss structure optimization problem to reduce weight[6]. Shim and Manoochehri minimized the material use subject to maximum stress constraints by a Simulated Annealing (SA) approach[7]. Besides these two popular methods, other algorithms have been investigated as well, such as ant colonies[8,9], particle swarms[10], harmony search[11], and bacterial foraging[12]. Non-gradient methods have four advantages over gradient-based methods[5]: better optima, applicable to discrete designs, free of gradients, and efficient to parallelize. However, the major disadvantage of the methods is their high computational cost from calling the objective functions, which becomes prohibitively expensive for large systems[3]. As a trade-off, sometimes searching space can be reduced in order for less computation. For instance, pattern search has been applied[13,14] which is a non-heuristic method with a smaller searching space but is more likely to be trapped in local minima.

Machine learning has been used in sequential model-based optimization (SMBO) to target at expensive objective function evaluation[15,16]. For instance, Bayesian optimization (BO)[17] uses a Gaussian prior to approximate the conditional probability distribution of an objective $p(y|x)$ where $y = F(x)$ is the objective and $x$ is the design variable (vector); then the unknown regions can be estimated by the probability model. In Covariance Matrix Adaptation Evolution Strategy (CMA-ES)[18], a multivariable Gaussian distribution is used to sample new queries. However, as demonstrated later in the paper, these methods are not designed for large-scale and high-dimensional problems, and thus do not perform well in topology optimization for slow convergence[19] or requirement of shrinking design space[20]. Despite some improvement to scale up these algorithms[21,22], none of them has shown superior performance in topology optimization to the best of our knowledge.

There are some reports on leveraging machine learning to reduce the computational cost of topology optimization[23–31]. Most of them are generative models which predict solutions of the same problem under different conditions, after being trained by optimized solutions from gradient-based methods. For example, Yu et al.[30] used 100,000 optimal solutions to a simple compliance problem with various boundary forces and the optimal mass fractions to train a neural network consisting of Convolutional Neural Network (CNN) and conditional Generative Adversarial Network (cGAN), which can predict near-optimal designs for any given boundary forces. However, generative models are not topology optimization algorithms: they rely on existing optimal designs as the training data. The predictions are restricted by the coverage of the training datasets. To consider different domain geometries or constraints, new datasets and networks would be required. Besides, the designs predicted by the networks are close to, but still different from the optimal designs. An offline learning method[31] replaces some FEM calculations during the optimization process with DNN's prediction, yet gives limited improvement especially considering that it requires the solutions to similar problems for training.

Here we propose an algorithm called Self-directed Online Learning Optimization (SOLO) to dramatically accelerate non-gradient topology optimization. A DNN is used to map designs to objectives as a surrogate model to approximate and replace the original function which is expensive to calculate. A heuristic optimization algorithm finds the possible optimal design according to DNN's prediction. Based on the optimum, new query points are dynamically generated and evaluated by the Finite Element Method (FEM) to serve as additional training data. The loop of such self-directed online learning is repeated until convergence. This iterative learning scheme, which can be categorized as an SMBO algorithm, takes advantage of the searching abilities of heuristic methods and the high computational speed of DNN. Theoretical convergence rate is derived under some assumptions. In contrast to gradient-based methods, this algorithm does not rely on gradient information of objective functions of the topology optimization problems. This property allows it to be applied to binary and discrete design variables in addition to continuous ones. To show its performance, we test the algorithm by two compliance minimization problems (designing solid so that the structure achieves maximum stiffness for a given loading), two fluid-structure optimization problems (designing fluid tunnel to minimize the fluid pressure loss for a given inlet speed), a heat transfer enhancement problem (designing a copper structure to reduce the charging time of a heat storage system), and three truss optimization problems (choosing the cross-sectional areas of bars in a truss). Our algorithm reduces the computational cost by at least two orders of magnitude compared with directly applying heuristic methods including Generalized Simulated Annealing (GSA), Binary Bat Algorithm (BBA), and Bat Algorithm (BA). It also outperforms an offline version (where all training data are randomly generated), BO, CMA-ES, and a recent algorithm based on reinforcement learning[32].

## Results

**Formulation and overview**. Consider the following topology optimization problem: in a design domain $\Omega$, find the material distribution $\rho(\mathbf{x})$ that could take either 0 (void) or 1 (solid) at point $\mathbf{x}$ to minimize the objective function $F$, subject to a volume constraint $G_0 \leq 0$ and possibly $M$ other constraints $G_j \leq 0$ $(j = 1, \ldots, M)$[4]. Mathematically, this problem can be written as looking for a function $\rho$ defined on the domain $\Omega$,

$$\min_{\text{dom}(\rho)=\Omega} F(\rho)$$
$$\begin{cases} G_0(\rho) = \int_\Omega \rho(\mathbf{x}) \, d\mathbf{x} - V_0 \leq 0 \\ G_j(\rho) \leq 0, \quad j = 1, \ldots, M \\ \rho(\mathbf{x}) = 0 \text{ or } 1, \quad \forall \mathbf{x} \in \Omega \end{cases} \quad (1)$$

where $V_0$ denotes the given volume. To solve such a problem numerically, the domain $\Omega$ is discretized into finite elements to

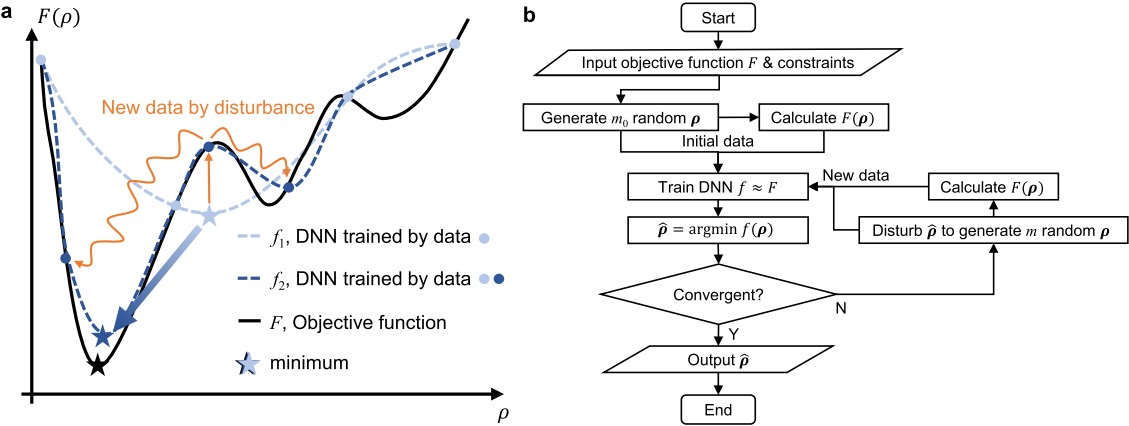

**Fig. 1 Schematics of the proposed self-directed online learning optimization. a** Schematic illustration of self-directed online training. The initial batch of training data (light-blue dots) is randomly located. The DNN $f_1$ (dashed light-blue line) trained on the first batch of data only gives a rough representation of the true objective function $F$ (solid black line). The second batch of training data (dark-blue dots) are generated by adding disturbance (orange curve) to the minimum of $f_1$. After trained with two batches, the DNN $f_2$ (dashed dark-blue line) is more refined around the minimum (the region of interest), while remains almost the same at other locations such as the right convex part. $f_2$ is very close to finding the exact global minimum point. **b** Flow diagram of the algorithm.

describe the density distribution by $N$ nodal or elemental values,

$$\min_{\boldsymbol{\rho}=[\rho_1,\rho_2,\cdots,\rho_N]^T} F(\boldsymbol{\rho})$$

$$\begin{cases} G_0(\boldsymbol{\rho}) = \sum_{i=1}^{N} w_i\rho_i - V_0 \leq 0 \\ G_j(\boldsymbol{\rho}) \leq 0, \quad j=1,\ldots,M \\ \rho_i \in S, \quad i=1,\ldots,N \end{cases} \quad (2)$$

where $w_i$ denotes the weight of integration. The domain of $\rho_i$ is usually binary ($S = \{0, 1\}$), but more generally may take other values such as discrete ($S = \{a_0, a_1, \ldots, a_K\}$) or continuous ($S = [0, 1]$).

Our algorithm can be applied to Eq. (2) with binary, discrete or continuous design variables. In this section, we discuss the case of continuous design variables since it is most general.

In many applications, the objective function is quite complicated and time-consuming to calculate, since it requires solving partial differential equations by, for instance, FEM. To reduce the number of FEM calculations and accelerate non-gradient optimization, we build a DNN to evaluate the objective function. In a naive way, the entire domain of the objective function should be explored to generate the training data. This would incur a huge number of FEM calculations. However, we only care about the function values close to the global optimum and do not require precise predictions in irrelevant regions. In other words, most information about the objective function in the domain is unnecessary except the details around the optimum. So we do not need to generate data to train in those irrelevant regions.

An intuitive explanation is shown in Fig. 1a. In a 1D minimization example, we can generate a small dataset to train the DNN and refine the mesh around the minimum obtained from the current prediction to achieve higher resolution at the place of interest in the next iteration. After several batches, the minimum of the predicted function would converge to that of the objective function.

Figure 1b shows the flow diagram of the proposed algorithm. A small batch of random vectors (or arrays) $\boldsymbol{\rho}$ satisfying the constraints in Eq. (2) is generated. The corresponding objective values $F(\boldsymbol{\rho})$ are calculated by FEM. Then, $\boldsymbol{\rho}$ and $F(\boldsymbol{\rho})$ are inputted into the DNN as the training data so that the DNN has a certain level of ability to predict the function values based on the design variables. Namely, the output of the DNN $f(\boldsymbol{\rho})$ approximates the

objective function $F(\boldsymbol{\rho})$. Next, the global minimum of the objective function $f(\boldsymbol{\rho})$ is calculated by a heuristic algorithm. After obtaining the optimized array $\hat{\boldsymbol{\rho}}$, more training data are generated accordingly. Inspired by the concept of GA[33], the disturbance we add to the array is more than a small perturbation, and is categorized as mutation, crossover, and convolution. Mutation means replacing one or several design variables with random numbers; crossover means exchanging several values in the array; convolution means applying a convolution filter to the variables (see "Methods" section for details). Then constraints are checked and enforced. The self-directed learning and optimization process stops when the value of the objective function $F(\hat{\boldsymbol{\rho}})$ does not change anymore or the computation budget is exhausted.

This algorithm can converge provably under some mild assumptions. Given the total number of training data $n_{\text{train}}$, for any trained DNN with small training error, we have

$$[F(\hat{\boldsymbol{\rho}}) - F^*]^2 \leq \tilde{O}\left(\frac{C}{\sqrt{n_{\text{train}}}}\right), \quad (3)$$

where $C$ is a constant related to some inherent properties of $F$ and DNN, $F^*$ is the global minimum of $F$, and $\tilde{O}$ omits log terms. This result states that when our trained DNN can fit the training data well, then our algorithm can converge to the global optimal value. We provide convergence guarantee with a concrete convergence rate for our proposed algorithm, and to the best of our knowledge, this is the first non-asymptotic convergence result for heuristic optimization methods using DNN as a surrogate model. The detailed theory and its derivation are elaborated in Supplementary Sect. 2.

In the following, we will apply the algorithm to eight classic examples of four types (covering binary, discrete and continuous variables): two compliance minimization problems, two fluid-structure optimization problems, a heat transfer enhancement problem, and three truss optimization problems.

**Compliance minimization.** We first test the algorithm on two simple continuous compliance minimization problems. We show that our algorithm can converge to the global optimum and is faster than other non-gradient methods.

As shown in Fig. 2a, a square domain is divided evenly by a $4 \times 4$ mesh. A force downward is applied at the top-right edge; the

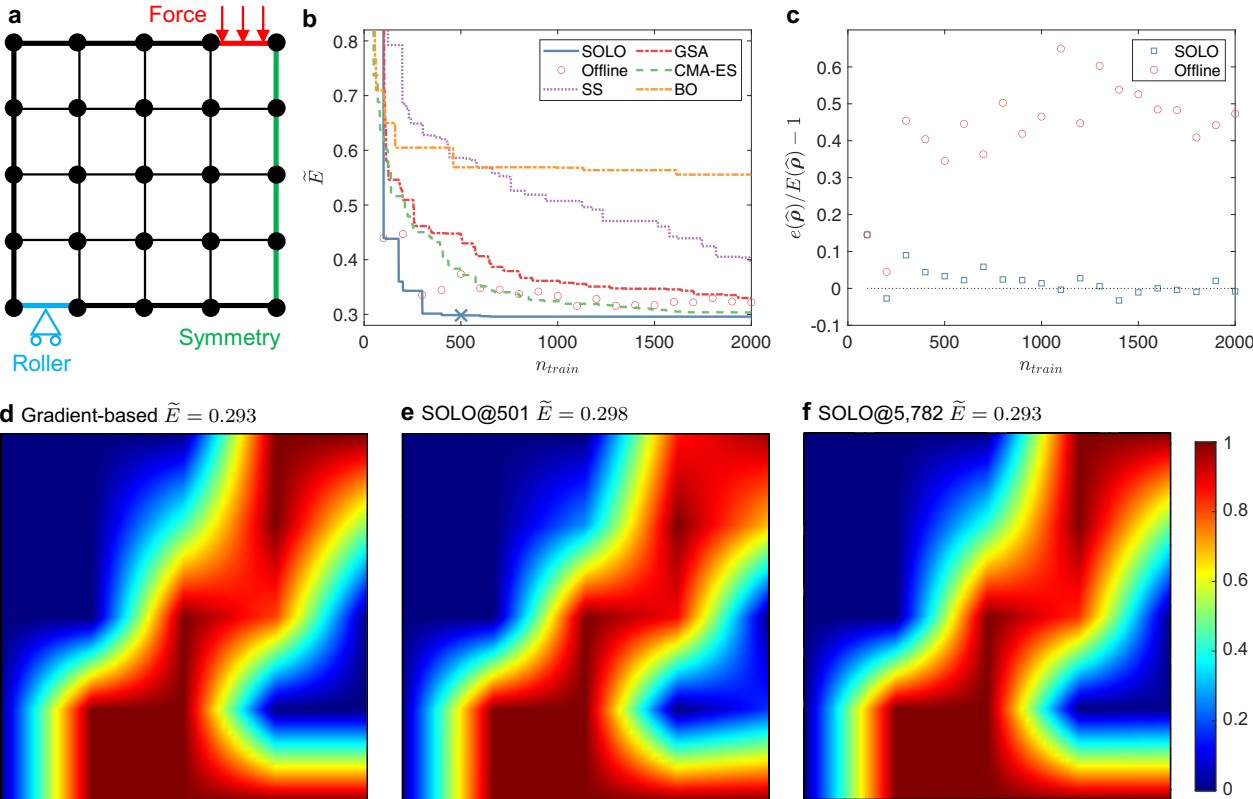

**Fig. 2 Setup and results of a compliance minimization problem with 5 × 5 design variables. a** Problem setup: minimizing compliance subject to a maximum volume constraint. **b** Best dimensionless energy with a total of $n_{train}$ accumulated training samples. SOLO denotes our proposed method where the cross "X" denotes the convergence point (presented in **e**), "Offline" denotes training a DNN offline and then uses GSA to search for the optimum without updating the DNN, whose results are independent so they are plotted as circles instead of a curve, SS denotes Stochastic Search, which is the same as SOLO except that $\hat{\rho}$ in each loop is obtained by the minimum of existing samples, CMA-ES denotes Covariance Matrix Adaptation Evolution Strategy, BO denotes Bayesian Optimization. SOLO converges the fastest among these methods. **c** Energy prediction error of $\hat{\rho}$ relative to FEM calculation of the same material distribution. $e(\hat{\rho})$ denotes DNN's prediction, $E(\hat{\rho})$ denotes FEM's result. **d** Optimized design produced by the gradient-based method. $\widetilde{E} = 0.293$. **e** Optimized design produced by SOLO. $n_{train} = 501$ and $\widetilde{E} = 0.298$. **f** Optimized design produced by SOLO. $n_{train} = 5782$ and $\widetilde{E} = 0.293$. In **d**–**f** dark red denotes $\rho = 1$ and dark blue denotes $\rho = 0$, as indicated by the right color scale bar.

bottom left edge is set as a roller (no vertical displacement); the right boundary is set to be symmetric. There are 25 nodal design variables to control the material distribution, i.e., density $\rho$. Our goal is to find the density $\rho_i (i = 1, 2, \ldots, 25)$, subject to a volume constraint of 0.5, such that the elastic energy $E$ of the structure is minimized, equivalent to minimizing compliance or the vertical displacement where the external force is applied. Formally,

$$\min_{\rho \in [0,1]^N} \widetilde{E}(\rho) = \frac{E(\rho)}{E(\rho_O)}, \quad (4)$$

where $\rho_O = [0.5, 0.5, \ldots, 0.5]^T$. The constraint is

$$\mathbf{w} \cdot \rho \leq 0.5, \quad (5)$$

where $\mathbf{w}$ denotes the vector of linear Gaussian quadrature. In Eq. (4), we use the dimensionless elastic energy $\widetilde{E}(\rho)$, defined as the ratio of elastic energy of the structure with any given material distribution to that of the reference uniform distribution (the material density is 0.5 everywhere in the domain). The elastic energy is calculated by FEM from the Young's modulus in the domain, which is related to density by the popular Simplified Isotropic Material with Penalization (SIMP) method [34],

$$Y(\rho(\mathbf{x})) = Y_0 \rho(\mathbf{x})^3 + \varepsilon \left[ 1 - \rho(\mathbf{x})^3 \right], \quad (6)$$

where $Y$ and $Y_0$ denote the Young's moduli as a variable and a constant, respectively, $\varepsilon$ is a small number to avoid numerical

singularity and $\rho(\mathbf{x})$ is the material density at a given location $\mathbf{x}$ interpolated linearly by the nodal values of the element.

For benchmark, we use a traditional gradient-based algorithm, the Method of Moving Asymptotes (MMA), to find the optimized solution (Fig. 2d).

For our proposed method, we use 100 random arrays to initialize the DNN. Then Generalized Simulated Annealing (GSA) is used to obtain the minimum $\hat{\rho}$ based on the DNN's prediction. Afterward, 100 additional samples will be generated by adding disturbance to $\hat{\rho}$ including mutation and crossover. Such a loop continues until convergence.

We compare our proposed method, Self-directed Online Learning Optimization (SOLO), with five other algorithms. In Fig. 2b, SOLO converges at $n_{train} = 501$. "Offline" denotes a naive implementation to couple DNN with GSA, which trains a DNN offline by $n_{train}$ random samples and then uses GSA to search for the optimum, without updating the DNN. As expected, the elastic energy decreases with the number of accumulated training samples $n_{train}$. This is because more training data will make the DNN estimate the elastic energy more accurately. Yet it converges much slower than SOLO and does not work well even with $n_{train} = 2000$. More results are shown in Supplementary Fig. 1. SS denotes Stochastic Search, which uses current minimum (the minimum of existing samples) to generate new searching samples; the setup is the same as SOLO except that the base design $\hat{\rho}$ is obtained from the current minimum instead of a

DNN. Comparing SS with SOLO, we can conclude that the DNN in SOLO gives a better searching direction than using existing optima. CMA-ES denotes Covariance Matrix Adaptation Evolution Strategy with a multivariable Gaussian prior. BO denotes Bayesian Optimization with Gaussian distribution as the prior and expected improvement (EI) as the acquisition function. Our method outperforms all these methods in terms of convergence speed. CMA-ES ranks the second with an objective value 3% higher than SOLO at $n_{train} = 2000$.

To assess inference accuracy in online and offline learning, we compare the DNN-predicted energy with that calculated by FEM on the same material distribution. The relative error is defined by $[e(\hat{\rho}) - E(\hat{\rho})]/E(\hat{\rho})$ where $e(\hat{\rho})$ and $E(\hat{\rho})$ denote energy calculated by DNN and FEM, respectively. The energy prediction error is shown in Fig. 2c. When $n_{train}$ is small, both networks overestimate the energy since their training datasets, composed of randomly distributed density values, correspond to higher energy. As $n_{train}$ increases, the error of SOLO fluctuates around zero since solutions with low energy are fed back to the network.

The solution of SOLO using 501 samples is presented in Fig. 2e, whose energy is 0.298, almost the same as that of the benchmark in Fig. 2d. With higher $n_{train}$, the solution from SOLO becomes closer to that of the benchmark (the evolution of optimized structures is shown in Supplementary Fig. 2). In Fig. 2f, the energy is the same as the benchmark. The material distribution in Fig. 2f does not differ much from that in Fig. 2e. In fact, using only 501 samples is sufficient for the online training to find the optimized material distribution. We find that in our problem and optimization setting, the GSA needs about $2 \times 10^5$ function evaluations to obtain the minimum of DNN. Since the DNN approximates the objective function, we estimate GSA needs the same number of evaluations when applying to the objective, then it means $2 \times 10^5$ FEM calculations are required if directly using GSA. From this perspective, SOLO reduces the number of FEM calculations to 1/400.

A similar problem with a finer mesh having 121 ($11 \times 11$) design variables is shown in Fig. 3a. The benchmark solution from MMA is shown in Fig. 3d, whose energy is 0.222. The trends in Fig. 3b, c are similar to those in Fig. 2 with a coarse mesh. Figure 3b shows that SOLO converges at $n_{train} = 10,243$, giving $\widetilde{E} = 0.228$. Our method again outperforms CMA-ES, the second-best algorithm according to Fig. 2b. The material distribution solutions are shown in Fig. 3e, f. The configuration of SOLO is the same as that for the coarse mesh except that each loop has 1000 incremental samples and GSA performs $4 \times 10^6$ function evaluations. Compared with directly using GSA, SOLO reduces the number of FEM calculations to 1/400 as well. The evolution of optimized structures is shown in Supplementary Fig. 3.

**Fluid-structure optimization.** In the following two problems, we leverage our algorithm to address binary fluid-structure optimization. We want to show that our method outperforms the gradient-based method and a recent algorithm based on reinforcement learning[32].

As shown in Fig. 4a, the fluid enters the left inlet at a given velocity perpendicular to the inlet, and flows through the channel bounded by walls to the outlet where the pressure is set as zero. In the $20 \times 8$ mesh, we add solid blocks to change the flow field to minimize the friction loss when the fluid flows through the channel. Namely, we want to minimize the normalized inlet pressure

$$\min_{\rho \in \{0,1\}^N} \widetilde{P}(\rho) = \frac{P(\rho)}{P(\rho_O)}, \qquad (7)$$

where $P$ denotes the average inlet pressure and $\rho_O = [0, 0, ..., 0]^T$

indicates no solid in the domain. As for the fluid properties, we select a configuration with a low Reynolds number for stable steady solution[35], specifically,

$$Re = \frac{DvL}{\mu} = 40, \qquad (8)$$

where $D$ denotes fluid density, $\mu$ denotes viscosity, $v$ denotes inlet velocity and $L$ denotes inlet width (green line).

For the benchmark, we use a typical gradient-based algorithm which adds an impermeable medium to change binary variables to continuous ones[36]. It uses the adjoint method to derive gradients and MMA as the solver. The solution is presented in Fig. 4c. The solid blocks form a ramp at the left bottom corner for a smooth flow expansion.

We use two variants of our algorithm. One is denoted as SOLO-G, a greedy version of SOLO where additional 10 samples produced in each loop are all from the DNN's prediction. The initial batch is composed of a solution filled with zeros and 160 solutions each of which has a single element equal to one and others equal to zero. The pressure values corresponding to these designs are calculated by FEM. These 161 samples are used to train a DNN. Next, Binary Bat Algorithm (BBA) is used to find the minimum of the DNN. The top 10 solutions (after removing repeated ones) encountered during BBA searching will be used as the next batch of training data. The other variant, denoted as SOLO-R, is a regular version of SOLO where each loop has 100 incremental samples. 10 of them are produced in the same way as SOLO-G whereas the rest 90 are generated by adding disturbance to the best solution predicted by the DNN. Similar to the compliance minimization problems, the disturbance includes mutation and crossover.

As shown in Fig. 4b, SOLO-G and SOLO-R converge to the same objective function value $\widetilde{P} = 0.9567$ at $n_{train} = 286$ and $n_{train} = 2148$, respectively. Their solutions are equivalent, shown in Fig. 4d and e. Intermediate solutions from SOLO-G are shown in Supplementary Fig. 4. We obtain the optimum better than the gradient-based method ($\widetilde{P} = 0.9569$) after only 286 FEM calculations. For comparison, a recent topology optimization work based on reinforcement learning used the same geometry setup and obtained the same solution as the gradient-based method after thousands of iterations[32]; our approach demonstrates better performance. Compared with directly using BBA which requires $10^8$ evaluations, SOLO-G reduces FEM calculations by orders of magnitude to about $1/(3 \times 10^5)$. To account for randomness, we repeat the experiments another four times and the results are similar to Fig. 4b (Supplementary Figs. 5 and 6).

We also apply our algorithm to a finer mesh, with $40 \times 16$ design variables (Fig. 5a). SOLO-G converges at $n_{train} = 1912$, shown in Fig. 5b. Our design (Fig. 5d, $\widetilde{P} = 0.8062$) is found to be better than the solution from the gradient-based algorithm (Fig. 5c, $\widetilde{P} = 0.8065$). Intermediate solutions from SOLO-G are shown in Supplementary Fig. 7. Compared with directly using BBA which needs $2 \times 10^8$ evaluations, SOLO-G reduces the number of FEM calculations to $1/10^5$. Similar trends can be observed when repeating the experiments (Supplementary Fig. 7). It is interesting to note that the optimum in Fig. 5d has two gaps at the 7th and 12th columns. It is a little counter-intuitive, since the gradient-based method gives a smooth ramp (Fig. 5c). We try filling the gaps and find that their existence indeed reduces pressure (Supplementary Fig. 8), which demonstrates how powerful our method is.

**Heat transfer enhancement.** In this example, we would like to solve a complicated problem that gradient-based methods are difficult to address. Phase change materials are used for energy storage by absorbing and releasing latent heat when the materials

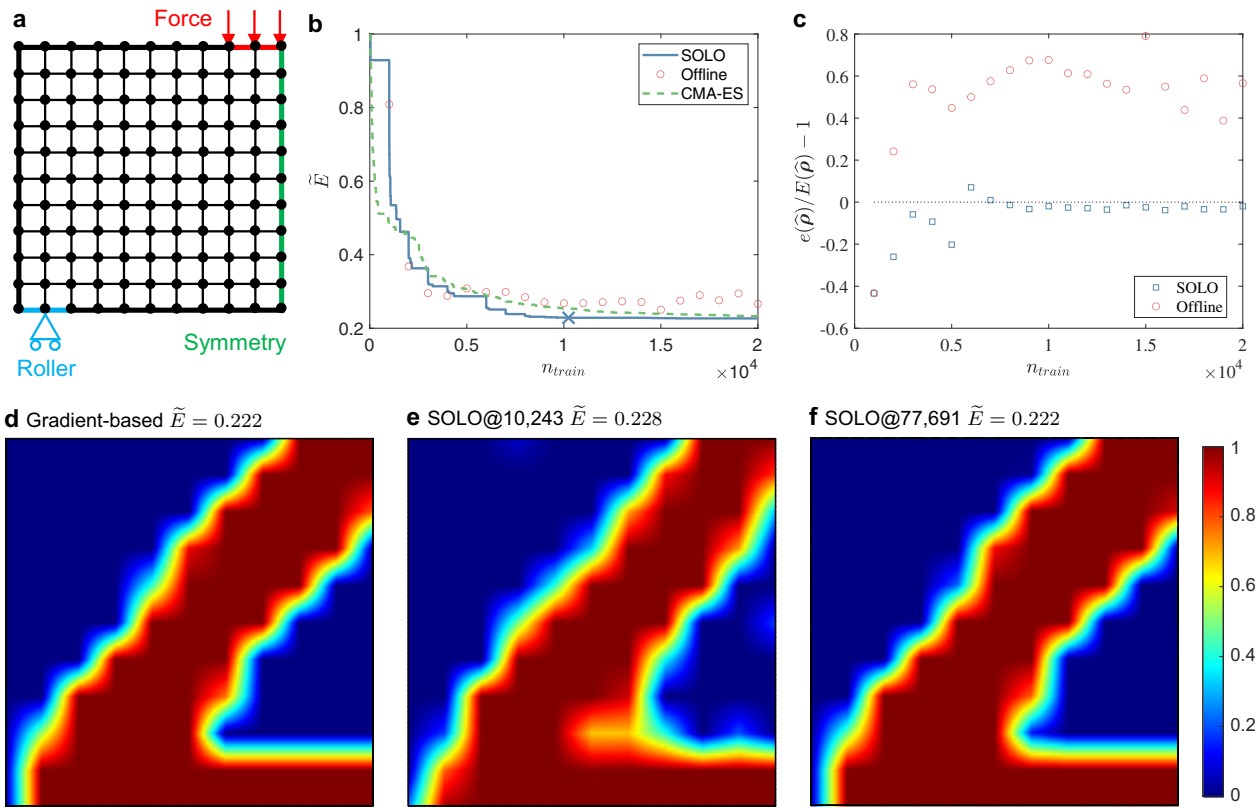

**Fig. 3 Setup and results of a compliance minimization problem with 11 × 11 design variables. a** Problem setup: minimizing compliance subject to maximum volume constraint. **b** Best dimensionless energy with a total of $n_{\text{train}}$ accumulated training samples. SOLO denotes our proposed method where the cross "X" denotes the convergence point (presented in **e**), "Offline" denotes training a DNN offline and then uses GSA to search for the optimum without updating the DNN, whose results are independent so they are plotted as circles instead of a curve, CMA-ES denotes Covariance Matrix Adaptation Evolution Strategy. SOLO converges the fastest among these methods. **c** Energy prediction error of $\hat{\rho}$ relative to FEM calculation of the same material distribution. $e(\hat{\rho})$ denotes DNN's prediction, $E(\hat{\rho})$ denotes FEM's result. **d** Optimized design produced by the gradient-based method. $\widetilde{E} = 0.222$. **e** Optimized design produced by SOLO. $n_{\text{train}} = 10{,}243$ and $\widetilde{E} = 0.228$. **f** Optimized design produced by SOLO. $n_{\text{train}} = 77{,}691$ and $\widetilde{E} = 0.222$. In **d–f** dark red denotes $\rho = 1$ and dark blue denotes $\rho = 0$, as indicated by the right bar.

change phases, typically between solid and liquid. Due to their simple structure and high heat storage capacity, they are widely used in desalination, buildings, refrigeration, solar system, electronic cooling, spacecraft and so forth[37]. However, commonly used non-metallic materials suffer from very low thermal conductivity. A popular solution is to add high conductivity material (such as copper) as fins to enhance heat transfer[38]. Topology optimization is implemented to optimize the geometry of fins. To deal with such transient problems, current gradient-based methods have to simplify the problem by using a predetermined time period and fixed boundary conditions[39–42]. By contrast, in real applications, these conditions depend on user demand and environment, or even couple with the temperature field of the energy storage system[43–47]. Therefore, problems with more complex settings need to be addressed.

We consider a heat absorption scenario where time is variant and the boundary condition is coupled with the temperature field. As shown in Fig. 6a, copper pipes containing heat source are inserted in a phase change material, paraffin wax RT54HC[48]; the heat source can be fast-charging batteries for electric vehicles or hot water for residential buildings. Considering symmetry, the problem is converted to a 2D problem in Fig. 6b. We fill the domain with wax to store heat and with copper to enhance heat transfer. The material distribution $\rho(\mathbf{x}) \in \{0, 1\}$ (1 being copper and 0 being wax) is represented by a $10 \times 10$ mesh. Specifically, a continuous function is interpolated by Gaussian basis functions

from the $10 \times 10$ design variables and then converted to binary values by a threshold (see Methods for details). Our goal is to find the optimal $\rho$ to minimize the time to charge the system with a given amount of heat

$$\min_{\rho \in [0,1]^N} \widetilde{t}(\rho) = \frac{t(\rho)}{t(\rho_O)}, \tag{9}$$

where $N = 100$, $\rho_O = [0, 0, \ldots, 0]^T$ means no copper inside the design domain, and $t(\rho)$ is the time to charge the system with $Q_0$ amount of heat, expressed by

$$\int_0^{t(\rho)} q(\rho, t)\mathrm{d}t = Q_0, \tag{10}$$

subject to the maximum heat flux constraint at the boundary (green curve in Fig. 6b)

$$q(\rho, t) \leq q_0, \tag{11}$$

the constraint of the maximum temperature of the domain,

$$T(\rho, q, \mathbf{x}, t) \leq T_0, \tag{12}$$

and given copper usage, i.e., the volume constraint of copper,

$$\frac{\int_\Omega \rho(\mathbf{x})\mathrm{d}\mathbf{x}}{\int_\Omega \mathrm{d}\mathbf{x}} = 0.2. \tag{13}$$

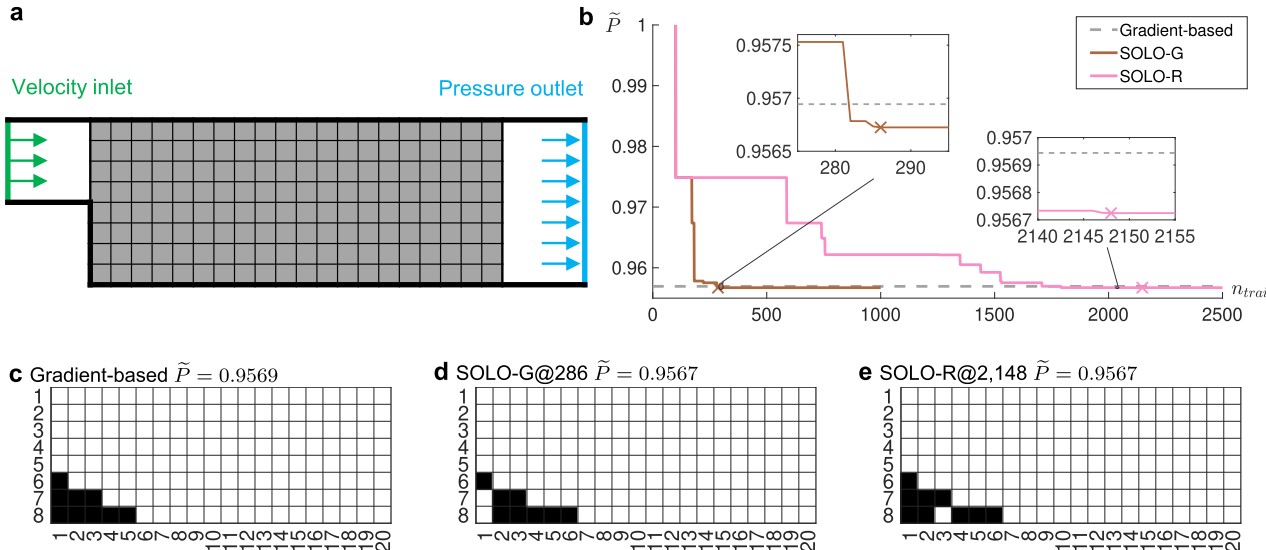

**Fig. 4 Setup and results of a fluid-structure optimization problem with 20 × 8 design variables. a** Problem setup: minimizing pressure drop through the tunnel. The vertical green line on the left denotes the inlet while the vertical blue line on the right denotes the outlet. **b** Dimensionless inlet pressure versus $n_{\text{train}}$, the number of accumulated training samples. SOLO-G denotes a greedy version of our proposed method, SOLO-R denotes the regular version of our proposed method. The horizontal dashed line denotes the solution from the gradient-based method. The cross "X" denotes the convergence point (presented in **d** and **e**, respectively). **c** Optimized design obtained by the gradient-based method. $\widetilde{P} = 0.9569$. **d** Optimized design obtained by SOLO-G. $n_{\text{train}} = 286$ and $\widetilde{P} = 0.9567$. **e** Optimized design obtained by SOLO-R. $n_{\text{train}} = 2148$ and $\widetilde{P} = 0.9567$. In **c**–**e** black denotes $\rho = 1$ (solid) and white denotes $\rho = 0$ (void). The solutions in **d** and **e** are equivalent since the flow is blocked by the black squares forming the ramp surface and the white squares within the ramp at the left bottom corner are irrelevant.

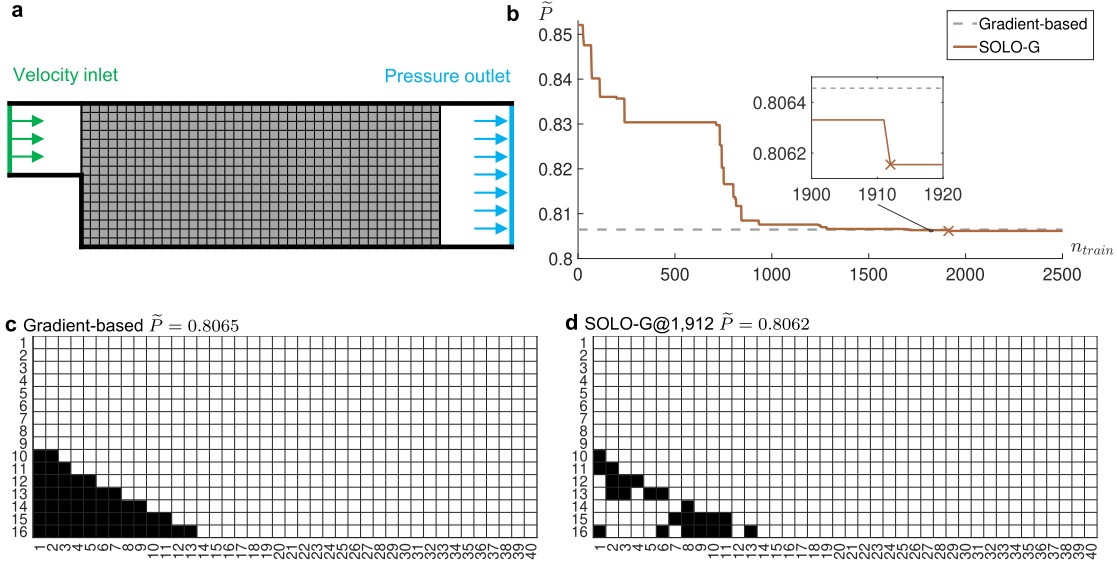

**Fig. 5 Setup and results of a fluid-structure optimization problem with 40 × 16 design variables. a** Problem setup: minimizing pressure drop through the tunnel. **b** Dimensionless inlet pressure versus $n_{\text{train}}$, the number of accumulated training samples. SOLO-G denotes a greedy version of our proposed method, where the cross "X" denotes the convergence point (presented in **d**). The horizontal dashed line denotes the solution from the gradient-based method. **c** Optimized design obtained by the gradient-based method. $\widetilde{P} = 0.8065$. **d** Optimized design obtained by SOLO-G. $n_{\text{train}} = 1912$ and $\widetilde{P} = 0.8062$. In **c**, **d** black denotes $\rho = 1$ (solid) and white denotes $\rho = 0$ (void). The SOLO-G result in **d** has two gaps at the 7th and 12th columns, while the gradient-based result in **c** gives a smooth ramp. We try filling the gaps and find that their existence indeed reduces pressure, which demonstrates the powerfulness of our method.

Here $Q_0$, $q_0$, and $T_0$ are preset constants. Obviously, the bottom left boundary (inner side of copper pipes) has the highest temperature during charging, thus we only need to consider the temperature constraint at this boundary. Physically, there are one or two charging steps: the system is charged at heat flux $q_0$ until the boundary temperature reaches $T_0$ or the total heat flow reaches $Q_0$ (whichever first), and if it is the former case, the heat flux is reduced to maintain the boundary temperature at $T_0$ until the total heat flow requirement is satisfied. In practice, we choose parameters such that the system will go through two steps for sure.

To solve the problem with objective Eq. (9) and constraints in Eqs. (11)–(13), our method SOLO is initialized by 500 random

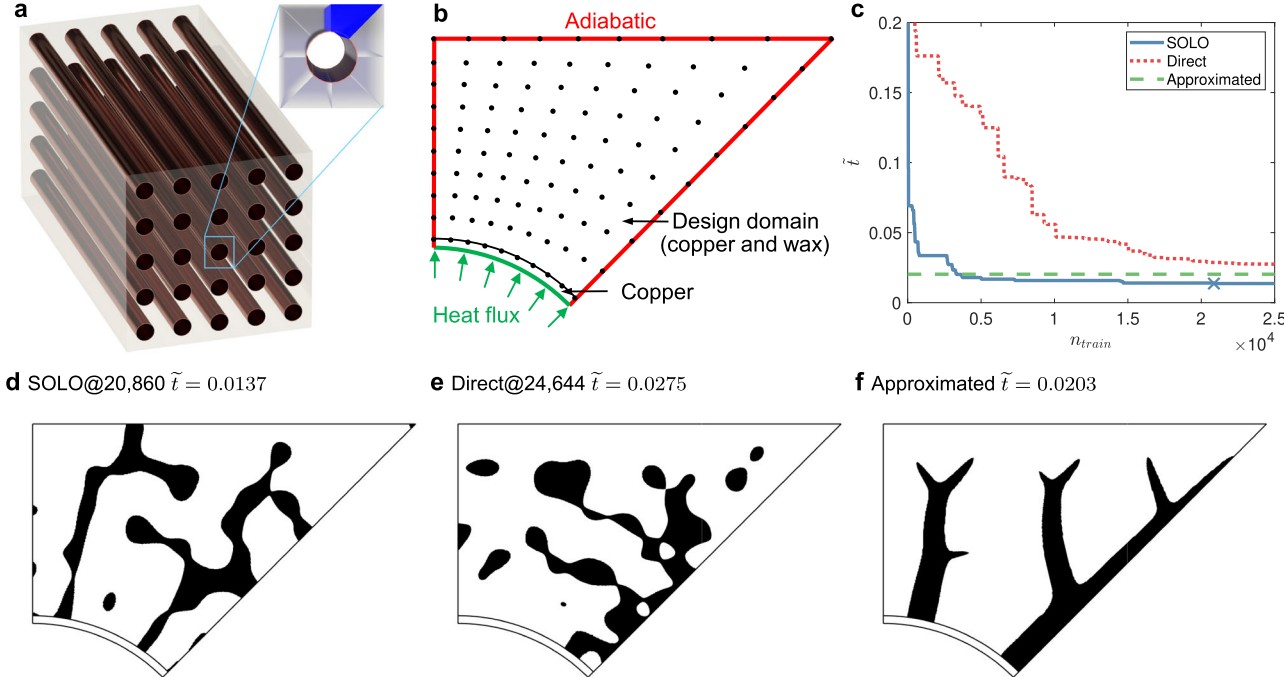

**Fig. 6 Setup and results of a heat transfer enhancement problem with 10 × 10 design variables. a** Engineering background: a group of copper pipes is inserted in a phase change material. Because of symmetry, we only need to consider 1/8 of the unit cell (dark-blue area in the top right corner). **b** Problem setup: minimizing the time to charge the system with a given amount of heat, subject to heat flux, temperature, and volume constraints. The black dots denote locations of design variables. **c** Dimensionless charging time versus $n_{train}$, the number of accumulated training samples. SOLO denotes our proposed method, where the cross "X" denotes the convergence point (presented in **d**). "Direct" denotes solving the problem directly by gradient descent. "Approximated" denotes simplifying this problem to a steady-state problem. **d** Optimized design obtained by SOLO. $\widetilde{t} = 0.0137$. **e** Optimized design obtained by "Direct". $n_{train} = 24,644$ and $\widetilde{t} = 0.0275$. **f** Optimized design obtained by "Approximated". $\widetilde{t} = 0.0203$. In **d**–**f** black denotes $\rho = 1$ (copper) and white denotes $\rho = 0$ (wax). The SOLO result in **d** has islands isolated from major branches, while the "Approximated" result in **f** gives a connected structure. We try combining the islands to be part of major branches and find that the existence of isolated islands indeed reduces time, which demonstrates the powerfulness of our method.

samples to train a DNN. Bat Algorithm (BA) is then used to find the minimum of the DNN, based on which additional 200 samples are generated in each loop by mutation and convolution. Two gradient-based methods are used as baselines to compare with our algorithm: one is to solve Problem (9)–(13) directly by gradient descent, denoted as "Direct"; the other is to simplify this problem to a steady-state problem[42], denoted as "Approximated". In Fig. 6c, SOLO converges at $n_{train} = 20,860$ (marked by a cross "X") with lower $\widetilde{t}$ than other methods. It appears counterintuitive that the solution of SOLO, shown in Fig. 6d, has some copper islands isolated from major branches. We tried removing these islands and adding more copper materials to the major branches to maintain copper volume, yet the variants showed worse performance, as shown in Supplementary Fig. 10. "Direct" gives the worst solution in Fig. 6e. "Approximated" yields a good solution with a tree structure, as shown in Fig. 6f; since it does not solve the same problem as the other two methods, we do not consider its relation with $n_{train}$ and represent it by a horizontal line in Fig. 6c.

Our method gives a good solution after 20,860 FEM calculations, while BA is estimated to need $4 \times 10^8$ calculations. In summary, our method outperforms the other two methods and reduces the number of FEM calculations by over four orders of magnitude compared with BA.

**Truss optimization.** In this example, we test the scalability of SOLO with over a thousand design variables. Also, we will compare it with a heuristic method, BA, to provide direct

evidence that SOLO can reduce the number of FEM computations by over two orders of magnitude.

Truss structures are widely used in bridges, towers, buildings, and so forth. An exemplary application, an antenna tower, is shown in Fig. 7a. Researchers have been working on optimizing truss structures from different perspectives. A classic truss optimization benchmark problem is to optimize a structure with 72 bars[49–52], as shown in Fig. 7b with four repeated blocks, so as to minimize the weight of the bars subject to displacement and tension constraint. Following this benchmark problem, we set the goal to optimize the size of each bar (the bars can all have different sizes) to minimize total dimensionless weight

$$\min_{\rho \in \{a_1, a_2, ..., a_{16}\}^N} \widetilde{W}(\boldsymbol{\rho}) = \frac{W(\boldsymbol{\rho})}{W(\boldsymbol{\rho}_{\max})} = \frac{\sum_{i=1}^N \rho_i L_i \gamma_i}{W(\boldsymbol{\rho}_{\max})}, \quad (14)$$

where $\rho_i$, $L_i$, and $\gamma_i$ are the cross-sectional area, length, and unit weight of the $i$th bar, respectively; $\boldsymbol{\rho}_{\max}$ uses the largest cross-sectional area for all bars; $N = 72$ is the number of bars. Each bar is only allowed to choose from 16 discrete cross-sectional area values $a_1, a_2, ..., a_{16}$, to represent standardized components in engineering applications. The tension constraint requires all bars to not exceed the maximum stress

$$|\sigma_i| \leq \sigma_0, \quad i = 1, 2, ..., N. \quad (15)$$

The displacement constraint is applied to the connections of the bars: the displacement in any direction is required to be lower

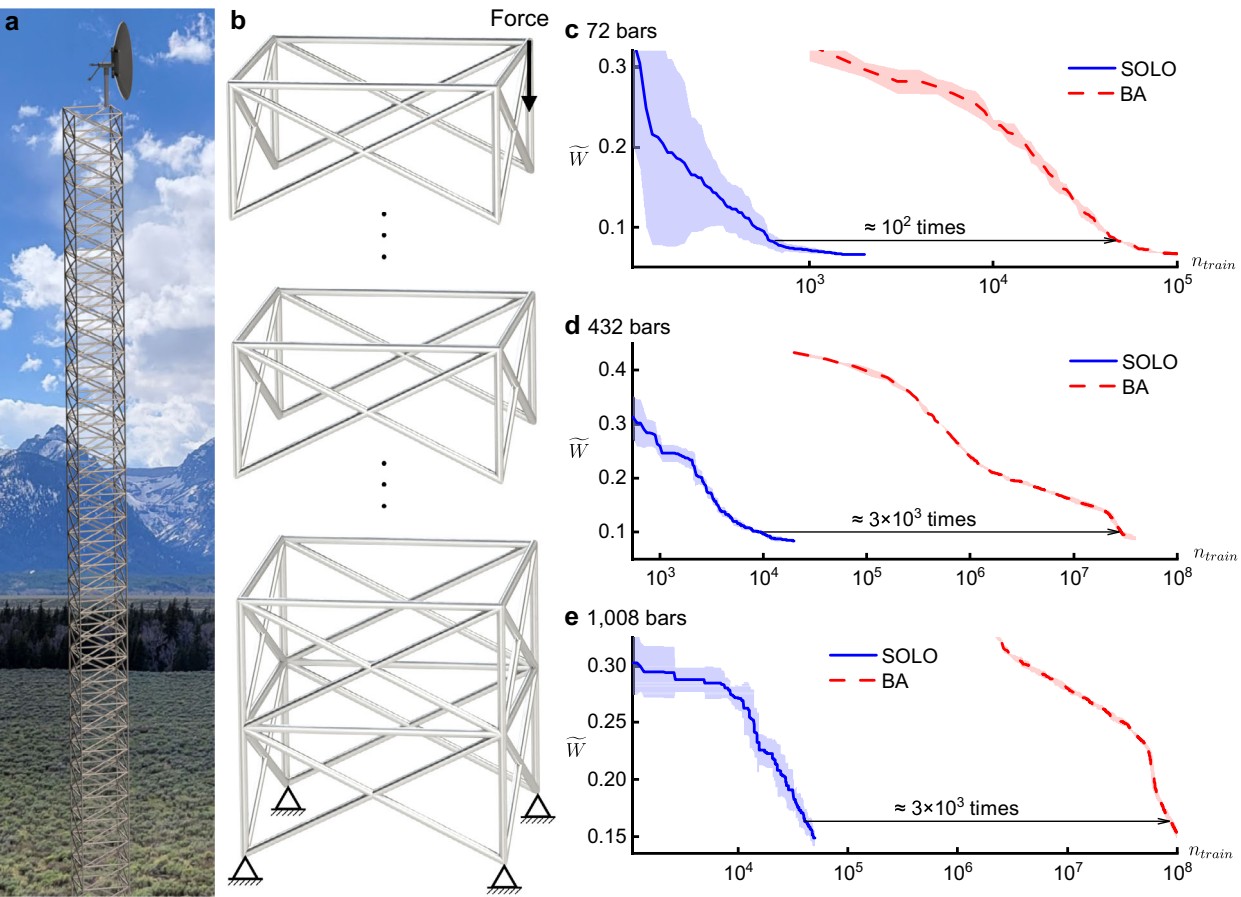

**Fig. 7 Setup and results of three truss optimization problems with different numbers of bars (equal to the numbers of design variables). a** Illustration of an antenna tower, an exemplary application of truss structures. **b** Illustration of the problem setup: minimizing total weight through changing the size of each bar, subject to stress and displacement constraints. The block is repeated until the given number of bars is reached. **c–e** Dimensionless weight $\widetilde{W}$ versus the number of accumulated training samples $n_{train}$. SOLO denotes our proposed method. BA denotes Bat Algorithm. The numbers of bars for these three sub-figures are 72, 432, and 1008, respectively. Each experiment is repeated five times; the curves denote the mean and the shadows denote the standard deviation.

than a threshold

$$||\Delta \mathbf{x}_i||_\infty \leq \delta_0, \quad i = 1, 2, ..., N_c, \quad (16)$$

where $N_c$ is the number of connections.

Now we have an optimization problem with objective Eq. (14) subject to stress constraint Eq. (15) and displacement constraint Eq. (16). In addition to the popular 72-bar problem, we add more repeated blocks to the structure to generate two more problems, with 432 and 1008 bars. Geometric symmetry is not considered while solving the problems. Therefore, the design space goes up to $16^{1008} \approx 10^{1214}$, which is extremely huge. For the three problems, SOLO is initialized by 100, 500, and 1000 samples, respectively. The number of incremental samples per loop is 10% of the initialization samples. 10% of incremental samples are the optima obtained by BA based on the DNN's prediction, and the rest 90% are generated by mutation of the best solution predicted by the DNN.

The results are shown in Fig. 7c–e. To reach the same objective weight, BA needs over $10^2$ times of calculations of SOLO. The difference becomes even larger when the number of variables increases. These examples demonstrate the scalability of SOLO by showing higher efficiency in computation, especially with a large number of design variables.

## Discussion

Topology optimization is an important problem with broad applications in many scientific and engineering disciplines. Solving non-linear high-dimensional optimization problems requires non-gradient methods, but the high computational cost is a major challenge. We proposed an approach of self-directed online learning optimization (SOLO) to dramatically accelerate the optimization process and make solving complex optimization problems possible.

We demonstrated the effectiveness of the approach in solving eight problems of four types, i.e., two compliance minimization problems, two fluid-structure optimization problems, a heat transfer enhancement problem, and three truss optimization problems. For the compliance problems with 25 and 121 continuous design variables, our approach converged and produced optimized solutions same as the known optima with only 501 and 10,243 FEM calculations, respectively, which are about 1/400 of directly using GSA and FEM without DNN based on our estimation. For the fluid problems with 160 and 640 binary variables, our method (SOLO-G) converged after 286 and 1912 FEM calculations, respectively, with solutions better than the benchmark. It used less than $1/10^5$ of FEM calculations compared with directly applying BBA to FEM, and converged much faster than another work based on reinforcement learning. In the heat transfer enhancement example, we investigated a

complicated, transient and non-linear problem. Our method gave a solution that outperformed other baselines after 20,860 FEM calculations, which was estimated to be four orders of magnitude less than BA. Similar to other SMBO methods, overhead computation was introduced (by training DNNs and finding their optima), but it was almost negligible (see the time profile in Supplementary Table 1) which is attractive for real-world applications where new designs want to be developed and tested. In these examples, we estimated the amount of computation of directly using heuristic algorithms, which showed that our approach led to 2–5 orders of magnitude of computation reduction. In addition to this estimation, we applied BA to the original objectives in the three truss optimization problems and observed 2–4 orders of magnitude of calculation reduction using our approach.

Our algorithm is neat and efficient, and has great potential for large-scale applications. We bring a new perspective for high-dimensional optimization by embedding deep learning in optimization methods. More techniques, such as parallel FEM computation, uncertainty modeling, and disturbance based on sensitivity analysis, can be incorporated to enhance the performance.

## Methods

**Enforcement of volume constraint**. Compliance and heat transfer problems have volume constraints. The latter will be detailed in Section Interpolation of design variables, thus we only discuss the former here. In the two compliance problems, all matrices representing the density distribution $\rho$ have the same weighted average $\sum_{i=1}^{N} w_i \rho_i = V_0$ due to the volume constraint where $w_i$ denotes the weight of linear Gaussian quadrature. A matrix from the initial batch is generated by three steps:

1. Generate a random matrix with elements uniformly distributed from 0 to 1.
2. Rescale the array to enforce the predefined weighted average.
3. Set the elements greater than one, if any, to one and then adjust those elements less than one to maintain the average.

Matrices for the second batch and afterward add random disturbance to optimized solutions $\hat{\rho}$ and then go through Step 2 and Step 3 above to make sure the volume satisfies the constraint.

**Finite Element Method (FEM) and gradient-based baselines**. The objective function values of material designs are calculated by FEM as the ground truth to train the DNN. In the compliance and fluid problems, the meshes of FEM are the same as the design variables. In the heat problem, the meshes are finer. Numerical results are obtained by COMSOL Multiphysics 5.4 (except the truss problems). Solutions from gradient-based methods (including "Approximated") are all solved by MMA via COMSOL with optimality tolerance as 0.001. In the fluid problems, the gradient-based baseline method produces a continuous array, and we use multiple thresholds to convert it to binary arrays and recompute their objective (pressure) to select the best binary array. In the heat problem, the "Approximated" method uses the same resolution as the other two methods (SOLO and "Direct") for a fair comparison. Specifically, we apply a Helmholtz filter[53], whose radius is half of the minimum distance of two design variable locations, to yield a mesh-independent solution. The solution is a continuous array; we use a threshold to convert it to a binary array which satisfies the volume constraint in Eq. (14).

**Interpolation of design variables**. In the two compliance problems and the heat problem, we use a vector (or matrix) $\rho$ to represent a spacial function $\rho(\mathbf{x})$. Interpolation is needed to obtain the function $\rho(\mathbf{x})$ for FEM and plotting. Given design variables $\rho = [\rho_1, \rho_2, ..., \rho_N]^T$, we get the values $\rho(\mathbf{x})$ by two interpolation methods. For the compliance problems, we use bilinear interpolation[54]. Suppose $\mathbf{x} = (x, y)$ is within a rectangular element whose nodal coordinates are $(x_1, y_1)$, $(x_1, y_2)$, $(x_2, y_1)$, $(x_2, y_2)$, the interpolated function value can be calculated by

$$\rho(x, y) = \frac{[x_2 - x \; x - x_1] \begin{bmatrix} F(x_1, y_1) & F(x_1, y_2) \\ F(x_2, y_1) & F(x_2, y_2) \end{bmatrix} \begin{bmatrix} y_2 - y \\ y - y_1 \end{bmatrix}}{(x_2 - x_1)(y_2 - y_1)}. \quad (17)$$

For the heat problem, a continuous function $\bar{\rho}(\mathbf{x}) \in [0, 1]$ (which will later be converted to a binary function which takes 0 or 1) is interpolated by Gaussian basis functions[13,20]:

$$\bar{\rho}(x, y) = \sum_{i=1}^{N} \lambda_i \phi(\mathbf{x}, \mathbf{x}_i) + a_0 + a_1 x + a_2 y, \quad (18)$$

where $\phi(\mathbf{x}, \mathbf{x}_i) = e^{-(\mathbf{x} - \mathbf{x}_i)^2 / d^2}$ ($d$ is a preset distance), and $\lambda_i, a_0, a_1, a_2$ are parameters to be determined. The following constraints are needed to guarantee a unique solution

$$\sum_{i=1}^{N} \lambda_i = 0, \sum_{i=1}^{N} \lambda_i x_i = 0, \sum_{i=1}^{N} \lambda_i y_i = 0. \quad (19)$$

Expressing the above equations by a matrix form, we have

$$\begin{bmatrix} \phi(\mathbf{x}_1, \mathbf{x}_1) & \cdots & \phi(\mathbf{x}_1, \mathbf{x}_N) & 1 & x_1 & y_1 \\ \vdots & \ddots & \vdots & \vdots & \vdots & \vdots \\ \phi(\mathbf{x}_N, \mathbf{x}_1) & \cdots & \phi(\mathbf{x}_N, \mathbf{x}_N) & 1 & x_N & y_N \\ 1 & \cdots & 1 & 0 & 0 & 0 \\ x_1 & \cdots & x_N & 0 & 0 & 0 \\ y_1 & \cdots & y_N & 0 & 0 & 0 \end{bmatrix} \begin{bmatrix} \lambda_1 \\ \vdots \\ \lambda_N \\ a_0 \\ a_1 \\ a_2 \end{bmatrix} = \begin{bmatrix} \rho_1 \\ \vdots \\ \rho_N \\ 0 \\ 0 \\ 0 \end{bmatrix}, \quad (20)$$

abbreviated as $\Phi \lambda = \begin{bmatrix} \rho \\ 0 \end{bmatrix}$. We get $\lambda = \Phi^{-1} \begin{bmatrix} \rho \\ 0 \end{bmatrix}$ and interpolate $\bar{\rho}(\mathbf{x})$ by Eq. (18). Then we set a threshold $\rho_{\text{thres}}$ to convert the continuous function $\bar{\rho}(\mathbf{x})$ to a binary one $\rho(\mathbf{x}) \in \{0, 1\}$, i.e., $\rho(\mathbf{x}) = 1$ if $\bar{\rho}(\mathbf{x}) \geq \rho_{\text{thres}}$ and $\rho(\mathbf{x}) = 0$ otherwise. The threshold $\rho_{\text{thres}}$ is controlled to satisfy the copper volume constraint Eq. (13).

**Deep Neural Network (DNN)**. The architectures of the DNN used in this paper are presented in Fig. 8. The design variable $\rho$ is flattened to a 1D vector as the input to DNN. All inputs are normalized before training and we introduce batch normalization (BN)[55] within the network as regularization. The output of DNN is reciprocal of the objective function (energy, pressure, charging time or weight) to give better resolution at lower objective values. For the rest of this paper, we regard the DNN to approximate the objective function for simplicity. To optimize the DNN training process, we apply ADAM[56] as the optimizer implemented on the platform of PyTorch 1.8.0[57]. The learning rate is 0.01. The loss function is set as Mean Square Error (MSE)[58]. All models are trained for 1000 epochs with a batch size of 1024 (if the number of training data is <1024, all the data will be used as one batch).

**Random generation of new samples from a base design**. After calculating the optimized array $\hat{\rho}$, more training data are generated by adding disturbance to it. As shown in Fig. 9, there are three kinds of disturbance: mutation, crossover, and convolution. They are all likely to change the weighted average of an array, thus the enforcement of volume constraint will be applied when necessary. Mutation means mutating several adjacent cells in the optimized array, i.e., generating random numbers from 0 to 1 to replace the original elements. In the 2D example shown in Fig. 9a, the numbers in a 2-by-2 box are set as random. Crossover denotes the crossover of cells in the array $\hat{\rho}$ and is achieved by the following steps:

1. Assign a linear index to each element in the array.
2. Randomly pick several indices.
3. Generate a random sequence of the indices.
4. Replace the original numbers according to the sequence above. As shown in Fig. 9b, indices are assigned sequentially from left to right and from top to bottom. The indices we pick in Step 2 are 3, 4, and 8; the sequence generated in Step 3 is 4, 8, and 3.

In the two compliance problems, the ways to generate a new input matrix based on $\hat{\rho}$ and their possibilities are:

(a) Mutation: mutating one element in $\hat{\rho}$ (10%);
(b) Mutation: mutating a $2 \times 2$ matrix in $\hat{\rho}$ (10%);
(c) Mutation: mutating a $3 \times 3$ matrix in $\hat{\rho}$ (20%);
(d) Mutation: mutating a $4 \times 4$ matrix in $\hat{\rho}$ (20%);
(e) Crossover: choosing an integer $n$ from one to the number of total elements, selecting $n$ cells in $\hat{\rho}$ and permuting them (20%);
(f) Generating a completely random matrix like the initial batch (20%).

In the fluid problem with $20 \times 8$ mesh, i.e., SOLO-R, the ways are the same as previous ones except a threshold is needed to convert the continuous array into a binary one. The threshold has a 50% probability to be $\beta_1^4$ where $\beta_1$ is uniformly sampled from $[0, 1]$, and has a 50% probability to be the element-wise mean of $\hat{\rho}$. In the heat problem, crossover is replaced by convolution. It is the same as the compliance problems except that (e) above is replaced by

(g) Convolution: substituting a submatrix of the array, whose size and the corresponding probability is the same as (a–d), with a same convolution (the output has the same size as the input submatrix) between the submatrix and $2 \times 2$ kernel whose element is uniformly sampled from $[0, 1]$.

In the truss optimization problems, the design variable $\rho$ is purely one-dimensional and can no longer be represented as a matrix. Therefore, we only use mutation. First, $\beta_2$ is uniformly sampled from $[0, 1]$ to indicate the ratio of elements to be mutated in $\hat{\rho}$, and then those elements are randomly selected to add $\gamma$ to

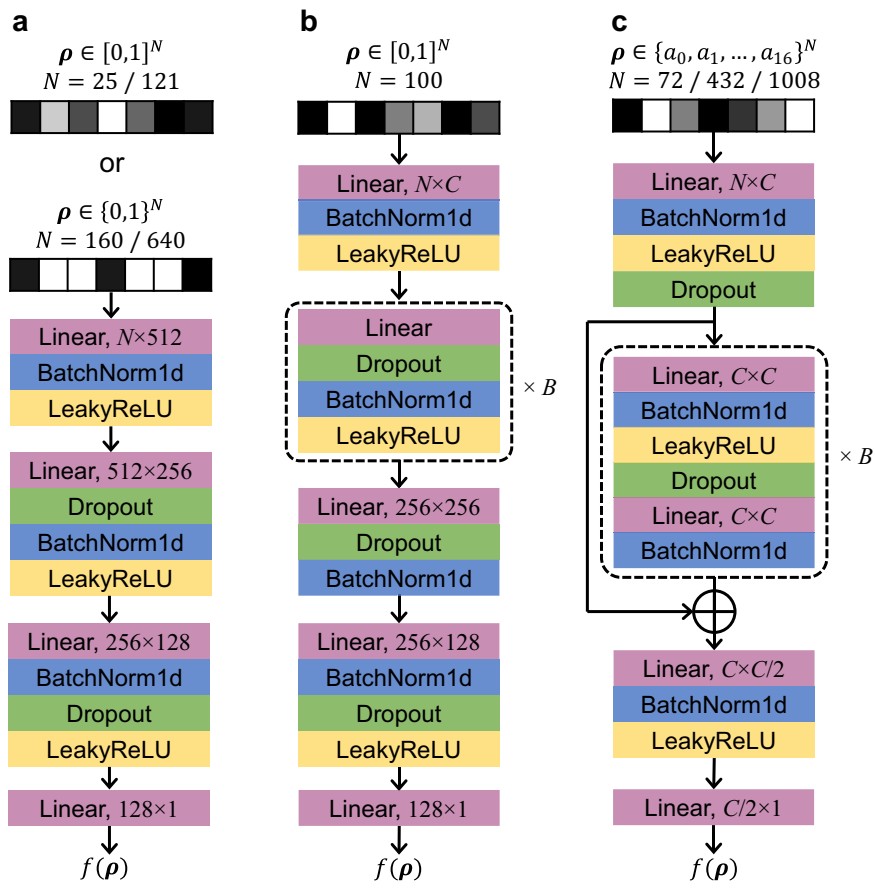

**Fig. 8 Architectures of DNN.** The input is a design vector $\rho$ and the output is the predicted objective function value $f(\rho)$. "Linear" presents a linear transformation and "BatchNorm1d" denotes one-dimensional batch normalization used to avoid internal covariate shift and gradient explosion for stable training[55]. "LeakyReLU" is an activation function extended from ReLU with activated negative values. "Dropout" is a regularization method to prevent overfitting by randomly masking nodes[68]. **a** The DNN in the compliance and fluid problems. **b** The DNN in the heat problem. Two architectures are used in this problem. At the 100th loop and before, $B = 1$, $C = 512$, and the Linear layer in the dashed box is $512 \times 256$. At the 101st loop and afterwards, $B = 4$, $C = 512$ and the 4 Linear layers are $256 \times 512$, $512 \times 512$, $512 \times 512$ and $512 \times 256$, respectively. **c** The DNN in the truss optimization problems. $B = 1$. $C = 512$ when $N = 72/432$; $C = 1024$ when $N = 1008$.

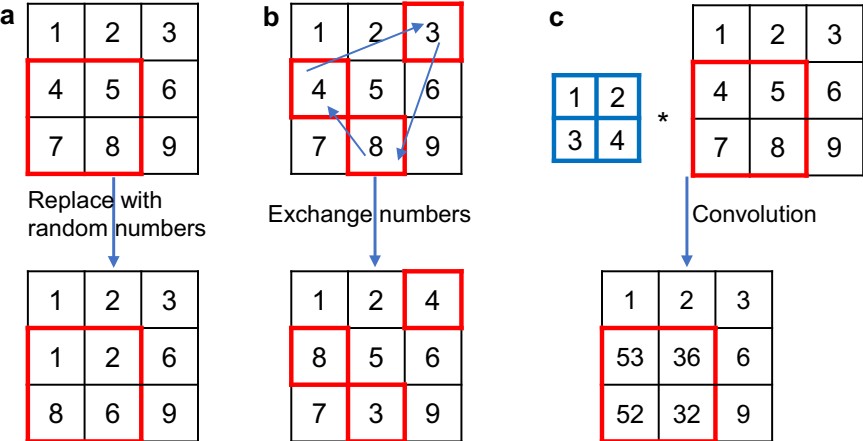

**Fig. 9 Illustration of mutation and crossover. a** An example of mutation: some adjacent cells (in the red box) are replaced with random numbers. **b** An example of crossover: several cells (in the red boxes) are exchanged. **c** An example of convolution: several cells (in the red box) are convoluted with a kernel (in the blue cell). The volume constraint may be enforced at the next step, not shown here.

themselves; $\gamma$ is uniformly sampled from $[-1, 1]$. Then the continuous variable is scaled and converted to the closest discrete one.

**Generalized Simulated Annealing (GSA).** Simulated Annealing (SA) is a stochastic method to determine the global minimum of a objective function by simulating the annealing process of a molten metal[59]. GSA is a type of SA with specific forms of visiting function and acceptance probability[60]. Assuming objective

$$\hat{\rho} = \arg\min_{\rho \in [0,1]^N} h(\rho),$$

(21)

we do the following:

1. Generate an initial state $\boldsymbol{\rho}^{(0)} = [\rho_1^{(0)}, \rho_2^{(0)}, ..., \rho_N^{(0)}]^T$ randomly and obtain its function value $E^{(0)} = h(\boldsymbol{\rho}^{(0)})$. Set parameters $T(0)$, $t_{\max}$, $q_v$, $q_a$.

2. For artificial time step $t = 1$ to $t_{\max}$,

   (a) Generate a new state $\boldsymbol{\rho}^{(t)} = \boldsymbol{\rho}^{(t-1)} + \Delta\boldsymbol{\rho}^{(t)}$, where the probability distribution of $\Delta\boldsymbol{\rho}^{(t)}$ follows the visiting function

$$g(\Delta\boldsymbol{\rho}^{(t)}) \propto \frac{[T(t)]^{-\frac{N}{3-q_v}}}{\left\{1 + (q_v - 1)\frac{[\Delta\rho^{(t)}]^2}{[T(t)]^{\frac{2}{3-q_v}}}\right\}^{\frac{1}{q_v-1}+\frac{N-1}{2}}}. \qquad (22)$$

   where $T$ denotes the artificial temperature calculated by

$$T(t) = T(0)\frac{2^{q_v-1} - 1}{(1+t)^{q_v-1} - 1}. \qquad (23)$$

   (b) Calculate the energy difference

$$\Delta E = E^{(t)} - E^{(t-1)} = h(\boldsymbol{\rho}^{(t)}) - h(\boldsymbol{\rho}^{(t-1)}). \qquad (24)$$

   (c) Calculate the probability to accept the new state

$$p = \min\left\{1, \left[1 - (1 - q_a)\frac{t}{T(t)}\Delta E\right]^{\frac{1}{1-q_a}}\right\}. \qquad (25)$$

   Determine whether to accept the new state based on the probability, if not, $\boldsymbol{\rho}^{(t)} = \boldsymbol{\rho}^{(t-1)}$.

3. Conduct a local search to refine the state.

Since compliance minimization has a volume constraint, the objective function used in the optimization process is written as

$$h(\boldsymbol{\rho}) = f(\boldsymbol{\rho}) + c(\mathbf{w} \cdot \boldsymbol{\rho} - V_0)^2, \qquad (26)$$

where $c$ is a constant to transform the constrained problem to an unconstrained problem by adding a penalty term. GSA is implemented via the SciPy package with default parameter setting. For more details, please refer to its documentation[61].

**Bat Algorithm (BA).** Bat Algorithm (BA) is a heuristic optimization algorithm, inspired by the echolocative behavior of bats[62]. This algorithm carries out the search process using artificial bats mimicking the natural pulse loudness, emission frequency, and velocity of real bats. It solves the problem

$$\hat{\boldsymbol{\rho}} = \arg\min_{\boldsymbol{\rho}\in[0,1]^N} h(\boldsymbol{\rho}). \qquad (27)$$

We adopt a modification[63] and implement as follows:

1. Generate $M$ vectors $\boldsymbol{\rho}^{(0,1)}, \boldsymbol{\rho}^{(0,2)}, ..., \boldsymbol{\rho}^{(0,M)}$. We use $\boldsymbol{\rho}^{(t,m)}$ to denote a vector, flattened from the array representing design variables. It is treated as the position of the $m$th artificial bat, where $m = 1, 2, ..., M$. We use $\rho_i^{(t,m)} \in [0, 1]$ to denote the $i$th dimension of vector $\boldsymbol{\rho}^{(t,m)}$, where $i = 1, 2, ..., N$. Thus, $\boldsymbol{\rho}^{(0,m)} = [\rho_1^{(0,m)}, \rho_2^{(0,m)}, ..\rho_N^{(0,m)}]^T$.

2. Calculate their function values and find the minimum $\boldsymbol{\rho}^* = \arg\min h(\boldsymbol{\rho}^{(0,m)})$.

3. Initialize their velocity $\mathbf{v}^{(0,1)}, \mathbf{v}^{(0,2)}, ..., \mathbf{v}^{(0,M)}, ..., \mathbf{v}^{(0,M)}$.

4. Determine parameters $q_{\min}$, $q_{\max}$, $t_{\max}$, $\alpha$, $\gamma$, $r^{(0)}$, $A^{(0)}$, $w_{\text{init}}$, $w_{\text{final}}$.

5. For artificial time step $t = 1$ to $t_{\max}$,

   (a) Update parameters $A^{(t)} = \alpha A^{(t-1)}$, $r^{(t)} = r^{(0)}(1 - e^{-\gamma t})$, $w^{(t)} = (1 - t/t_{\max})^2(w_{\text{init}} - w_{\text{final}}) + w_{\text{final}}$.

   (b) For $m = 1, 2, ..., M$,

      i. Calculate sound frequency

$$q^{(t,m)} = q_{\min} + (q_{\max} - q_{\min})\beta, \qquad (28)$$

      where $\beta$ is a random number that has a uniform distribution in $[0, 1]$.

      ii. Update velocity based on frequency

$$\mathbf{v}^{(t,m)} = w^{(t)}\mathbf{v}^{(t-1,m)} + (\boldsymbol{\rho}^{(t-1,m)} - \boldsymbol{\rho}^*)q^{(t,m)}. \qquad (29)$$

      iii. Get a (temporary) new solution. Calculate the new position

$$\boldsymbol{\rho}^{(t,m)} = \boldsymbol{\rho}^{(t,m-1)} + \mathbf{v}^{(t,m)}. \qquad (30)$$

      iv. Local search. Generate $\beta_i'(i = 1, 2, ..., N)$, a series of random numbers uniformly sampled in $[0, 1]$. For those $i$ satisfying $\beta_i' > r^{(t)}$, add noise to the current best solution

$$\rho_i^{(t,m)} = \rho_i^* + \epsilon A^{(t)}, \qquad (31)$$

where $\epsilon$ is a random variable sampled in Gaussian distribution with zero mean, $\rho_i^*$ is the $i$th component of $\boldsymbol{\rho}^*$. If $\rho_i^{(t,m)}$ goes over the range $[0, 1]$, it is thresholded to 0 or 1. For others, keep them as they are.

      v. Determine whether to accept the new solution. Reverse to the previous step $\boldsymbol{\rho}^{(t,m)} = \boldsymbol{\rho}^{(t-1,m)}$, if $h(\boldsymbol{\rho}^{(t,m)}) > h(\boldsymbol{\rho}^{(t-1,m)})$ or $\beta'' > A^{(t)}$ (where $\beta''$ is random number uniformly sampled in $[0, 1]$).

   (c) Update $\boldsymbol{\rho}^* = \arg\min_{m=1,2,...,M} h(\boldsymbol{\rho}^{(t,m)})$.

6. Output $\hat{\boldsymbol{\rho}} = \boldsymbol{\rho}^*$.

BA is used in the heat and truss problems. In the heat problem, we optimize $f$ without adding penalty terms since the volume constraint is controlled by a threshold, i.e., $h = f$. In the truss optimization problems, we need to choose $\boldsymbol{\rho}^{(t,m)}$ in a discrete space since only 16 values are allowed. Before we evaluate $h(\boldsymbol{\rho}^{(t,m)})$, we will replace $\rho_i^{(t,m)}$ by the nearest discrete values. To deal with constraints in Eqs. (15) and (16), the objective function is converted to

$$h(\boldsymbol{\rho}) = W(\boldsymbol{\rho})\left(1 + \sum_{|\sigma_i| > \sigma_0}\frac{|\sigma_i| - \sigma_0}{\sigma_0} + \sum_{||\Delta\mathbf{x}_i||_\infty > \delta_0}\frac{||\Delta\mathbf{x}_i||_\infty - \delta_0}{\delta_0}\right)^2. \qquad (32)$$

**Binary Bat Algorithm (BBA).** Binary Bat Algorithm[64,65] is a binary version of BA. To solve

$$\hat{\boldsymbol{\rho}} = \arg\min_{\boldsymbol{\rho}\in\{0,1\}^N} h(\boldsymbol{\rho}), \qquad (33)$$

we slightly adjust the original algorithm and implement it as follows:

1. Generate $M$ vectors $\boldsymbol{\rho}^{(0,1)}, \boldsymbol{\rho}^{(0,2)}, ..., \boldsymbol{\rho}^{(0,M)}$. We use $\boldsymbol{\rho}^{(t,m)}$ to denote a vector, flattened from the array representing design variables. It is treated as the position of the $m$th artificial bat, where $m = 1, 2, ..., M$. We use $\rho_i^{(t,m)} \in \{0, 1\}$ to denote the $i$th dimension of vector $\boldsymbol{\rho}^{(t,m)}$, where $i = 1, 2, ..., N$. Thus, $\boldsymbol{\rho}^{(0,m)} = [\rho_1^{(0,m)}, \rho_2^{(0,m)}, ..\rho_N^{(0,m)}]^T$.

2. Calculate their function values and find the minimum $\boldsymbol{\rho}^* = \arg\min h(\boldsymbol{\rho}^{(0,m)})$.

3. Initialize their velocity $\mathbf{v}^{(0,1)}, \mathbf{v}^{(0,2)}, ..., \mathbf{v}^{(0,M)}, ..., \mathbf{v}^{(0,M)}$.

4. Determine parameters $q_{\min}$, $q_{\max}$, $t_{\max}$, $\alpha$, $\gamma$, $r^{(0)}$, $A^{(0)}$.

5. For artificial time step $t = 1$ to $t_{\max}$,

   (a) Update parameters $A^{(t)} = \alpha A^{(t-1)}$, $r^{(t)} = r^{(0)}(1 - e^{-\gamma t})$.

   (b) For $m = 1, 2, ..., M$,

      i. Calculate sound frequency

$$q^{(t,m)} = q_{\min} + (q_{\max} - q_{\min})\beta, \qquad (34)$$

      where $\beta$ is a random number that has a uniform distribution in $[0, 1]$.

      ii. Update velocity based on frequency

$$\mathbf{v}^{(t,m)} = \mathbf{v}^{(t-1,m)} + (\boldsymbol{\rho}^{(t-1,m)} - \boldsymbol{\rho}^*)q^{(t,m)}. \qquad (35)$$

      iii. Get a (temporary) new solution. Calculate the possibility to change position based on velocity

$$V_i^{(t,m)} = \left|\frac{2}{\pi}\arctan\left(\frac{\pi}{2}v_i^{(t,m)}\right)\right| + \frac{1}{N}. \qquad (36)$$

      iv. Random flip. Generate $\beta_i'(i = 1, 2, ..., N)$, a series of random numbers uniformly in $[0, 1]$. For those $i$ satisfying $\beta_i' < V_i^{(t,m)}$, change the position by flipping the 0/1 values

$$\rho_i^{(t,m)} = 1 - \rho_i^{(t-1,m)}. \qquad (37)$$

      For others, keep them as they are.

      v. Accept the local optimum. Generate $\beta_i''(i = 1, 2, ..., N)$, a series of random numbers uniformly sampled in $[0, 1]$. For those $i$ satisfying $\beta_i'' > r^{(t)}$, set $\rho_i^{(t,m)} = \rho_i^*$.

      vi. Determine whether to accept the new solution. Reverse to the previous step $\boldsymbol{\rho}^{(t,m)} = \boldsymbol{\rho}^{(t-1,m)}$, if $h(\boldsymbol{\rho}^{(t,m)}) > h(\boldsymbol{\rho}^{(t-1,m)})$ or $\beta''' > A^{(t)}$ (where $\beta'''$ is random number uniformly sampled in $[0, 1]$).

   (c) Update $\boldsymbol{\rho}^* = \arg\min_{m=1,2,...,M} h(\boldsymbol{\rho}^{(t,m)})$.

6. Output $\hat{\boldsymbol{\rho}} = \boldsymbol{\rho}^*$.

BBA is used in the fluid problems. Since we do not have constraints in these problems, we can optimize $f$ without adding penalty terms, i.e., $h = f$.

## Data availability

The optimization data generated in this study have been deposited in the Zenodo database[66].

## Code availability

All code (MATLAB and Python) used in this paper is deposited in the Zenodo repository[67] or available at https://github.com/deng-cy/deep_learning_topology_opt.

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

## Acknowledgements

This work was supported by the National Science Foundation under Grant No. CNS-1446117 (W.L.).

## Author contributions

C.D. designed the algorithm and drafted the manuscript. Y.W. derived the convergence theory. C.D. and C.Q. wrote the code. Y.W., C.Q., and Y.F. edited the manuscript. W.L. conceived this work, supervised the study, and revised the manuscript.

## Competing interests

The authors declare no competing interests.
