## [Peer Review File · Nature Communications]

Reviewers' comments:

Reviewer #1 (Remarks to the Author):

This paper proposes a self-directed online learning method for topology optimization. My main concerns are:

1. The proposed scheme shows the superiority over an offline trained neural network. However, the contrastive network is only trained on some randomly generated samples, which is not fair in the comparison. The dataset used in the training should be discussed in detail, which is very important for the offline trained models.
2. In line 256, authors claimed the proposed approach is over 100 times faster than pre-trained DNN. However, offline learning usually only needs one-time training. The training time should not be included in topology optimization because the offline trained models are only used for inference.
3. Deep learning approaches have been widely used in topology optimization. There are also unsupervised learning approaches to address the dataset issue. Please include them in the examples and results session.
4. The majority of the current topology optimization software utilize pure density-based methods (e.g. SIMP or BESO) or hybrid approaches. However, this paper only shows the speed improvement over the GSA approach. I suggest to include the other widely used approaches for a comprehensive comparison. Moreover, it is not clear why GSA needs about 2×10^5 FEM calculations for a compliance minimization problem with 25 design variables.
5. The paper only shows two examples. More cases should be provided to show the robustness of the proposed scheme. Moreover, it only shows the prediction error relative to FEM calculation of the same material distribution. The accuracy between the proposed scheme and the GSA should also be provided.
6. Authors claimed all models are trained for 1000 epochs with a batch size of 1024. The number of batches is not clear.
7. In line 72, GSA stands for Generalized Simulated Annealing. However, in line 331, it changes to Generative Simulated Annealing.

Reviewer #2 (Remarks to the Author):

Even though I consider the basic idea worth further investigations, the current manuscript lacks a thorough discussion of the assumptions underlying the approach, their implications, as well as a more meaningful comparison to existing approaches.

First of all, I was somehow surprised about the statement "Because of the intrinsic limitation of gradient-based algorithms, the majority of existing approaches have only been applied to simple compliance minimization problems since they would fail as soon as the problem becomes complicated such as involving varying signs on gradients or non-linear constraints."

For instance, as shown in the work by Wu et al. (2016), A system for high-resolution topology optimization, the design domains shown in the current manuscript can be simulated easily in less than a second using gradient-based optimization. In addition, it is possible to simulate much larger and even 3D domains, which doesn't seem to be easily possible using the proposed approach. To make a fair comparison, it is further mandatory to specify the residuals up to which the solutions have been computed, which has not been done in the current work. I also recommend to more carefully distinguish between the time required by FEM-analysis to compute the compliance, and the gradient-based optimization step.

It is further said "Stochastic methods have four advantages over gradient-based methods: better

optima, applicable to discrete designs, free of gradients and efficient to parallelize”

What is meant by “better optima”? An optima is an optima, do you mean finding a better local optima than gradient-based approaches can find? I don’t see this is confirmed by the current experiments. Furthermore, isn’t it also true that your current optimization problem is a non-discrete one? And finally, also FE-based approaches can be parallelized effectively – see in particular current work on GPU parallelization. SA parallelisation is often done by domain splitting or combinatorial optimization. Both seem problematic in the current application scenario, since they can affect the optimized topology.

It is said “In many applications, the objective function is quite complicated and time-consuming for calculations, since it requires solving partial differential equations by, for instance, FEM. To accelerate computation, we build a DNN to evaluate the objective function.” Do you mean acceleration of FEM analysis or gradient-based optimization?

Further, “In traditional machine learning, the entire domain of the objective function should be explored to generate the training data. This would incur huge amount of FEM calculations. However, we only care about the function values close to the global optimum and do not require precise predictions in irrelevant regions.”

A couple of thoughts here: When training a network, you can, in principle, let the network learn to progressively move from a current topology to the updated one, by considering external parameters and domain-specific parameters. Learning would probably be done using localized sub-domains, to learn – if possible – characteristic material deposition patterns that occur repeatedly throughout the domain. Then, you would not have to consider irrelevant regions. On the other hand, it often occurs in topology optimization that empty regions become populated with increasing optimization iterations. This situation may become a problematic case, and I think it is also problematic for your proposed approach.

It is said “After obtaining the optimized array pbase, more training data are generated nearby.”

What does “nearby” mean here? If one knows that the solution is nearby, also gradient-based approach can be employed effectively. Then it is said that mutation and crossover is used to disturb the density arrays. Isn’t it so that such disturbances can results in significant changes in the objective function, and in particular ill-posed topologies? How does this affect the prediction of the compliance?

Overall, since training is embedded into the simulation process, I’m wondering about the performance of the approach when more complicated and especially 3D domains are used. This is especially because a fully convolutional network is used, which becomes extremely expensive for higher resolutions and dimensions. Furthermore, since the probabilistic sampling is only around the current solution, how can the system step into vastly different configurations, which in particular occur at the beginning of the iterative optimization process?

It is said that the DNN is used to estimate $F(p)$ instead of solving the differential equations. I do recommend to first demonstrate this in a separate study. I believe this would be a significant result on its own, yet needs further analysis. It is not clear to me that a mapping from physical material parameters, boundary conditions, and a given domain to the compliance can be easily learned. If this is possible, then I would first try to embed network-based inference into classical gradient-based optimization schemes, since the optimization process requires only about 10%-15% of the overall time.

Summary evaluation: In my opinion, the manuscript addresses a very interesting and important topic in topology optimization, i.e., to predict the compliance of a given topology using CNNs and to search for a locally optimal solution using SA. However, these are two different aspects which need to be

analysed separately before they can be combined as demonstrated. The analysis, in my opinion, does not confirm the claims made by the authors. The description is often vague, and a more thorough experimental analysis - using in particular different design domains - is missing. A more elaborate comparison to state-of-the-art solvers using gradient-based optimization is mandatory.

Reviewer #3 (Remarks to the Author):

This paper proposes a self-directed learning method with DNN for topology optimization. It is an online learning approach that generates training samples dynamically around the optimum with the finite element method. Experiments on two compliance problems show that this approach is effective and efficient compared to offline learning baselines. Overall, I think the novelty of this approach is limited.

The strengths of this paper are: (1) addressing the topology optimization problem with deep neural networks seems to be interesting. (2) experiments show that the proposed approach is efficient.

The weakness of this paper: (1) some related works are missing, i.e., 3D Topology Optimization using Convolutional Neural Networks; Deep Generative Design: Integration of Topology Optimization and Generative Models, etc. The authors shall spend more effort in justifying the novelty of the method. (2) The experiment on the two examples is not enough to account for the claim that it can solve the optimization problems for large systems; The systems demonstrated is not complicated enough. (3) Model complexity analysis is expected instead of just empirical running time comparison. (4) The writing and organization can be improved.

Overall, I tend to reject this paper.

Response to the Reviewer’s Comments

Dear editors and reviewers:

Thank you for your careful reviews and suggestions to our manuscript. We have studied the comments and made major changes in the manuscript. We have addressed all comments in the revised manuscript. All revised/new parts were highlighted in red the manuscript. Following the reviewers’ suggestions, we comprehensively revised our manuscript (including about 50% new materials). We also created a completely new 17-page Supplementary Information (including theoretical proof).

Reviewer #1 (Remarks to the Author):

1. The proposed scheme shows the superiority over an offline trained neural network. However, the contrastive network is only trained on some randomly generated samples, which is not fair in the comparison. The dataset used in the training should be discussed in detail, which is very important for the offline trained models.

Response:

There might be some misunderstanding on our definition of “online” and “offline” methods. Here the major difference between online and offline is the training data (i.e., the online method generates training data smartly). The comparison is to show how this “dynamic” data is better. So this difference in data is exactly what we want to compare.

The online scheme uses the optimal solution from the previous iteration to generate new training data. The new data is random but is based on the previous optimal solution. Offline scheme means all training samples are purely random, all generated at once prior to training. We understand that conventionally the words “online” and “offline” are often used to describe the method to feed data to DNN with the same dataset, with “online” denoting feeding data during training and “offline” denoting using the whole dataset to train. Our definition in this paper is motivated by this concept but highlights the use of different dataset. We believe the impact from the difference in how to train the network is very minor. What matters is the dataset, as the reviewer has mentioned. Our paper emphasizes that the DNN in the online scheme goes through training and inference iteratively (thus we call online), and we want to compare the method to randomly generate data without inference (which we call offline).

To more clearly define “Offline”, we state in our paper:

(Introduction, Line 73) **...offline version (where all training data are randomly generated)...**

(Examples and results, Line 155) **“Offline” denotes a naive implementation to couple DNN with GSA, which trains a DNN offline by n_{train} random samples and then uses GSA to search for the optimum, without updating the DNN.**

(Caption of Fig.2) **“Offline” denotes training a DNN offline and then uses GSA to search for the optimum without updating the DNN.**

2. In line 256, authors claimed the proposed approach is over 100 times faster than pre-trained DNN. However, offline learning usually only needs one-time training. The training time should not be included in topology optimization because the offline trained models are only used for inference.

Response:

We agree with the reviewer that “Offline” only needs one-time training. Compared with Offline, our approach introduces overhead, which is about twice of FEM computation time. Considering this overhead, we need to show that “Offline” needs 300 times more training samples to safely say the proposed approach is over 100 times faster. We verified this claim and showed in Supplementary Figure 1 (copied below). As can be seen, “Offline” is still not satisfactory at $n_{train} = 2 \times 10^5$, which is about 400 times of the 500 samples required by the online method (which we denoted as SOLO). Since “Offline” is not as good as “Online” with even 400 times of data, we can safely claim that the proposed approach is over 100 times faster than pre-trained DNN.

We want to note that we choose to emphasize the comparison of our approach with state-of-the-art methods which are better than “Offline”, so we delete this statement in the manuscript to de-emphasize “Offline”.

3. Deep learning approaches have been widely used in topology optimization. There are also unsupervised learning approaches to address the dataset issue. Please include them in the examples and results session.

Response:

We searched the literature on topology optimization where unsupervised learning approaches are used. We found two types of methods that can possibly be categorized as unsupervised learning: one is generative methods, and the other is reinforcement learning. Generative methods aim to increase data, but they are not dealing with the problem that we are looking at, since 1) they need optimized solutions beforehand to train the network. Our approach does not need to know the

optimized solution. 2) they normally use more than 10,000 samples to train a network, which is higher than what we use. We added an introduction of generative methods: (Introduction, Line 47) “There are some reports on leveraging machine learning to reduce the computational cost of topology optimization^{19–25}. Generative models are used to...”

For reinforcement learning, we added comparison between our approach and a reinforcement learning approach published in the most recent year for a fluid-structure optimization problem. We added a comparison (Page 11, Line 226) “We obtain the optimum better than the gradient-based method ($\tilde{P} = 0.9569$) after only 286 FEM calculations. For comparison, a recent topology optimization work based on reinforcement learning used the same geometry setup and obtained the same solution as the gradient-based method after thousands of iterations; our approach demonstrates better performance.”

We are happy to explore comparison with more methods if the reviewer is referring to any other unsupervised learning approaches.

4. The majority of the current topology optimization software utilize pure density-based methods (e.g. SIMP or BESO) or hybrid approaches. However, this paper only shows the speed improvement over the GSA approach. I suggest to include the other widely used approaches for a comprehensive comparison. Moreover, it is not clear why GSA needs about 2×10^5 FEM calculations for a compliance minimization problem with 25 design variables.

Response:

We thank the reviewer for the suggestion. We added comparison with several other gradient-free methods in the two compliance minimization examples, where we used SIMP as the benchmark. Besides, we added two fluid-structure optimization problems where we compare our method with the adjoint method and a new algorithm based on reinforcement learning. Please see pages 6-11 of the manuscript.

GSA needs about 2×10^5 FEM calculations for a compliance minimization problem with 25 design variables is based on our observation. This is not a large number. Imagine if we search by a grid fashion, even with only 2 grids for each variable, this will need $2^{25} = 3 \times 10^7$ FEM calculations. GSA is already much more efficient.

As for how to get the number, we modify the description of the manuscript: (Line 182) “We find that in our problem and optimization setting, the GSA needs about 2×10^5 function evaluations to obtain the minimum of DNN. Since the DNN approximates the objective function, we estimate GSA needs the same number of evaluations when applying to the objective, then it means 2×10^5 FEM calculations are required if directly using GSA.”

5. The paper only shows two examples. More cases should be provided to show the robustness of the proposed scheme. Moreover, it only shows the prediction error relative to FEM calculation of the same material distribution. The accuracy between the proposed scheme and the GSA should also be provided.

Response:

We thank the reviewer for the valuable suggestion. We added more methods and more examples, as shown in pages 6-11 of the manuscript.

Here accuracy is to evaluate the prediction of the DNN relative to the ground truth (FEM). Following the reviewer's suggestion, we also added a comparison between our approach and directly applying GSA. We showed that our solution is better than GSA by giving a lower energy.

6. Authors claimed all models are trained for 1000 epochs with a batch size of 1024. The number of batches is not clear.

Response:

The number of batches is not a constant since the training data in our approach is not a constant. So epoch and batch size are better descriptors. One epoch means going through all the data once. For instance, if we train on $m=10240$ samples with $b=1024$ batch size, then one epoch means $m/b=10$ batches.

If the reviewer is talking about the case where the number of training data is less than the batch size (i.e. $m/b < 1$), we add a note to clarify:

(Methods-Structure of DNN, Line 296) "...with a batch size of 1,024 (if the number of training data is less than 1,024, all the data will be used as one batch)."

7. In line 72, GSA stands for Generalized Simulated Annealing. However, in line 331, it changes to Generative Simulated Annealing.

Response:

Thanks for pointing out and sorry for the typo. GSA stands for Generalized Simulated Annealing. We have corrected this mistake.

Reviewer #2 (Remarks to the Author):

1. *Even though I consider the basic idea worth further investigations, the current manuscript lacks a thorough discussion of the assumptions underlying the approach, their implications, as well as a more meaningful comparison to existing approaches.*

Response:

We thank the reviewer for the suggestion. We have added two parts in our revised manuscript.

1) Theoretical proof. Regarding to assumptions and implications, we mathematically proved that our approach has a guaranteed convergence rate as (Eq.(4))

$$(F(\hat{\rho}) - F^*)^2 \leq \tilde{O}\left(\frac{C}{\sqrt{n_{train}}}\right),$$

where n_{train} is the total number of training data C is a constant related to some inherent properties of F and DNN, F^* is the global minimum of F , and \tilde{O} omits log terms. A detailed theorem is shown in Supplementary Section 2:

$$\left(F(\hat{\rho}^{(T)}) - F(\rho^*)\right)^2 \leq 4C(T) \left(\frac{96B^2}{\sqrt{mT}} \sqrt{d^2 D \log(1 + 8BB_W^D D \sqrt{mTd})} + 12B^2 \sqrt{\frac{2\log^2 \delta}{mT} + \frac{8}{mT}} + \epsilon \right).$$

2) Comparison. We added a comparison with state-of-the-art methods in two continuous problems. Besides, we added two binary problems and show that our approach outperforms the gradient-based method and a recent work based on reinforcement learning. These new results and materials are given in pages 6-11 of the manuscript.

2. *First of all, I was somehow surprised about the statement “Because of the intrinsic limitation of gradient-based algorithms, the majority of existing approaches have only been applied to simple compliance minimization problems since they would fail as soon as the problem becomes complicated such as involving varying signs on gradients or non-linear constraints.” For instance, as shown in the work by Wu et al. (2016), A system for high-resolution topology optimization, the design domains shown in the current manuscript can be simulated easily in less than a second using gradient-based optimization. In addition, it is possible to simulate much larger and even 3D domains, which doesn’t seem to be easily possible using the proposed approach. To make a fair comparison, it is further mandatory to specify the residuals up to which the solutions have been computed, which has not been done in the current work. I also recommend to more carefully distinguish between the time required by FEM-analysis to compute the compliance, and the gradient-based optimization step.*

Response:

This statement was not generated by us, but rather a rephrase from Ref [4] in the manuscript.

The original sentence in reference [4] reads “For simple compliance minimization problems with a single volume constraint, basically any rigorous mathematical approach as well as intuition-based schemes will work fine and produce nice plots of material distributions in limited number of iterations...As soon as more difficult problems like non-selfadjoint problems with varying signs on gradients or problems involving more than one, possibly non-linear, constraint are considered, the majority of the intuition-based approaches fail. If the goal is to work with a wide palette of multiphysics optimization problems with non-trivial and multiple constraints, the number of applicable optimizers becomes very limited.”

In our opinion, the word “simple” here does not refer to the number of design variables in a problem. Rather, it denotes the nature of a problem, i.e., governing equations and constraints. We understand the statement may be too strong, following the reviewer’s comment we revised it to:

(Line 22) **Because of the intrinsic limitation of gradient-based algorithms, the majority of existing approaches have only been applied to simple problems, since they would fail as soon as the problem becomes complicated such as involving varying signs on gradients or non-linear constraints.**

Our proposed method can be used for both 2D and 3D problems with no difference in terms of implementation flow, and the limiting factor is only the number of design variables (grids).

For residuals, we discuss both gradient and gradient-free methods below. As for gradient residuals, we added the details of our implementation of the gradient-based methods (Methods-Finite Element Method (FEM), Line 281) “**Numerical results are obtained by COMSOL Multiphysics 5.4. Solutions from gradient-based methods are also obtained by COMSOL with optimality tolerance as 0.001.**” For gradient-free methods, there is no similar concept like “residuals”. Considering that one objective (energy or pressure in our examples) may correspond to different designs, we use the achieved values of objective functions for comparing different methods.

Our method is gradient-free. As explained below, we compare the solutions, not the computational time of our method with that of the gradient-based methods; nor do we accelerate them. Therefore, there is no need to decouple FEM-analysis and gradient-based updating step. Below we elaborate more on our logic to compare with gradient-based methods. In the two compliance minimization problems, we use the gradient-based solution as the true optimal solution, and compare different (gradient-free) approaches to show that ours can converge to the global optimum the fastest. In the newly added fluid problems, we do not know the true optimal solution, thus we compare different approaches (including gradient-based and gradient-free). Still, when we compare with the gradient-based method, we did not compare convergence speed or computation time, but only the outputted solution. In these two problems, we outperform the gradient-based methods by giving better solutions. The focus of this manuscript is to develop a gradient-free method. Please also see the response to question 9 (the last question from the reviewer), which discusses the relation between our methods and gradient-based methods.

[4] Maute, K. and Sigmund, O., 2013. Topology optimization approaches: A comparative review. *Structural and Multidisciplinary Optimization*, 6.

3. It is further said “Stochastic methods have four advantages over gradient-based methods: better optima, applicable to discrete designs, free of gradients and efficient to parallelize”. What is meant

by “better optima”? An optima is an optima, do you mean finding a better local optima than gradient-based approaches can find? I don’t see this is confirmed by the current experiments. Furthermore, isn’t it also true that your current optimization problem is a non-discrete one? And finally, also FE-based approaches can be parallelized effectively – see in particular current work on GPU parallelization. SA parallelisation is often done by domain splitting or combinatorial optimization. Both seem problematic in the current application scenario, since they can affect the optimized topology.

Response:

These sentences in our introduction are used to review the gradient-free methods in general. “better optima” was a term we followed in a review paper Ref [12], “NGTO (non-gradient topology optimization) uses global search and hence converges to better optima than local search GTO (gradient topology optimization).” As the reviewer has mentioned, the meaning of “better optima” in this manuscript is finding a better optimum than gradient-based approaches. Our current optimization problem was a non-discrete one. To address these concerns, we add two discrete problems where our method finds better solution than a gradient-based method. In general, gradient-free methods have more flexibility and will not be trapped in local optima. Thus, we say gradient-free methods can find better optima.

Similarly, “efficient to parallelize” was rephrased from the review paper Ref [12], “NGTO runs efficiently and scales perfectly on parallel computers.” Although there have been attempts to parallelize FEM, SA and other algorithms, dividing a task into dependent sub-tasks will always be worse than directly handing independent tasks, since CPU/GPU cores have to communicate if tasks are dependent (Amdahl's law). We believe it is safe to claim many non-gradient approaches (which use independent calculations) are more efficient in terms of parallelization.

[12] Sigmund, O., 2011. On the usefulness of non-gradient approaches in topology optimization. Structural and Multidisciplinary Optimization, 43(5), pp.589-596.

4. It is said “In many applications, the objective function is quite complicated and time-consuming for calculations, since it requires solving partial differential equations by, for instance, FEM. To accelerate computation, we build a DNN to evaluate the objective function.” Do you mean acceleration of FEM analysis or gradient-based optimization?

Response:

Neither, we do not accelerate FEM itself or gradient-based optimization (our approach is a gradient-free optimization). We want to reduce the number of FEM calculations needed to obtain the optimum of an objective by leveraging DNN. Following the reviewer’s question, we revised the sentence to (Line 89) “To reduce the number of FEM calculations and accelerate gradient-free optimization, we build a DNN to evaluate the objective function.”

5. Further, “In traditional machine learning, the entire domain of the objective function should be explored to generate the training data. This would incur huge amount of FEM calculations. However, we only care about the function values close to the global optimum and do not require precise predictions in irrelevant regions.” A couple of thoughts here: When training a network, you can, in principle, let the network learn to progressively move from a current topology to the

updated one, by considering external parameters and domain-specific parameters. Learning would probably be done using localized sub-domains, to learn – if possible – characteristic material deposition patterns that occur repeatedly throughout the domain. Then, you would not have to consider irrelevant regions. On the other hand, it often occurs in topology optimization that empty regions become populated with increasing optimization iterations. This situation may become a problematic case, and I think it is also problematic for your proposed approach.

Response:

The intuition of our algorithm is to search both globally and locally. It is possible some undesired designs (empty regions are populated, or ill-posed topologies) are queried by the algorithm. In our implementation, we give a high objective to penalize such designs, so that the probability of such designs will be lower in the following iterations due to the output of DNN’s prediction. In other words, our approach can handle the kind of problematic cases that the reviewer mentioned, so that it is not an issue. This was confirmed in our application examples.

6. It is said “After obtaining the optimized array pbase, more training data are generated nearby.” What does “nearby” mean here? If one knows that the solution is nearby, also gradient-based approach can be employed effectively. Then it is said that mutation and crossover is used to disturb the density arrays. Isn’t it so that such disturbances can results in significant changes in the objective function, and in particular ill-posed topologies? How does this affect the prediction of the complacance?

Response: Here the distance described by “nearby” is not in the sense of Euclidean space, instead we mean the points that can be obtained by mutation or crossover. To avoid confusion, we change the word to “accordingly”:

(Line 108) “...more training data are generated accordingly.”

We agree with the reviewer that such disturbances can result in significant changes in the objective function. As we replied to the previous comment, since we are using SIMP, a high objective will be outputted to penalize ill-posed topologies. We want the DNN to learn from them, so that the probability of such designs will be lower in the next iterations. Another solution is to limit the trials within well-posed topologies, which is not in the scope of this manuscript but can be added to our approach.

7. Overall, since training is embedded into the simulation process, I’m wondering about the performance of the approach when more complicated and especially 3D domains are used. This is especially because a fully convolutional network is used, which becomes extremely expensive for higher resolutions and dimensions. Furthermore, since the probabilistic sampling is only around the current solution, how can the system step into vastly different configurations, which in particular occur at the beginning of the iterative optimization process?

Response:

First we would like to explain about convolutional networks (CNNs). We did not use CNNs as our network; instead, we only use fully connected layers. CNNs are commonly used in vision models

for the equivariance of the convolution operation, in another word, local features are independent of positions. Yet, we do not assume such properties in the topological designs, although in practice they may be good enough to learn the function, which requires further study.

Then for the high dimension issue, we agree with the reviewer that the problem becomes more difficult to solve for higher dimensions. Right now we formulate variables separately and do not have any prior assumption on the properties. Such problems are NP-hard problems which do not have a solver with polynomial complexity. Currently the maximum dimension in our manuscript is 640, and we could envision the difficulty when the dimension goes higher (e.g., >10,000). At this stage, the approach is hard to directly apply to higher dimensions, but it is the state-of-the-art gradient-free method in such a scenario, and has great potential to extend. For further studies, some dimension reduction techniques can be used, such as discarding some dimensions according to sensitivity analysis, and grouping several variables to form a certain pattern.

As for global convergence, we have shown a mathematical proof. The intuitive explanation is that the probabilistic sampling is not only around the current solution; the disturbance may be huge to dramatically change the configurations: even in the worst case where $\hat{\rho}$ is trapped in a local minimum, the disturbance is able to generate designs allowing the DNN to shift its prediction to another configuration.

8. It is said that the DNN is used to estimate $F(\rho)$ instead of solving the differential equations. I do recommend to first demonstrate this in a separate study. I believe this would be a significant result on its own, yet needs further analysis. It is not clear to me that a mapping from physical material parameters, boundary conditions, and a given domain to the compliance can be easily learned. If this is possible, then I would first try to embed network-based inference into classical gradient-based optimization schemes, since the optimization process requires only about 10%-15% of the overall time.

It has been proved that a DNN with ReLU activation function can approximate any function in Sobolev space (bounded norm and bounded gradient) ^[*]. The upper bound of generalization error of a DNN has been investigated ^[†]. The studies show that, with enough training data and small training error, any complicated function can be learned by DNN. The mapping can be learned, but probably not easily. If the material parameters and boundary conditions are all variables, a huge number of training data is required. Therefore, it is inefficient for our gradient-free method to explore the whole domain to learn such a mapping, and we choose a smart strategy to learn the mapping only at places of interest. We believe it is more inefficient for a gradient-based method to learn the whole domain mapping, unless the mapping will be reused.

[*] Yarotsky, D., 2017. Error bounds for approximations with deep ReLU networks. *Neural Networks*, 94, pp.103-114.

[†] Attias, I., Kontorovich, A. and Mansour, Y., 2019, March. Improved generalization bounds for robust learning. In *Algorithmic Learning Theory* (pp. 162-183). PMLR.

9. Summary evaluation: In my opinion, the manuscript addresses a very interesting and important topic in topology optimization, i.e., to predict the compliance of a given topology using CNNs and to search for a locally optimal solution using SA. However, these are two different aspects which

need to be analysed separately before they can be combined as demonstrated. The analysis, in my opinion, does not confirm the claims made by the authors. The description is often vague, and a more thorough experimental analysis - using in particular different design domains - is missing. A more elaborate comparison to state-of-the-art solvers using gradient-based optimization is mandatory.

Response:

We thank the reviewer for the valuable suggestions. We put significant effort into these and added theoretical analysis of the convergence rate, more examples and more comparisons. Moreover, we added testing of our approach on a different design domain for the binary fluid-structure problems.

We appreciate the reviewer’s advice on more comparison with state-of-the-art gradient-based solvers. We added comparison of our method with a gradient-based solver in the two binary problems. Further, we focus more on comparison with gradient-free solvers, and added comparison with several gradient-free methods such as CMA-ES, BO, and a RL-based algorithm published recently. In our opinion, as an initial work of the algorithm, we want to test more on problems that have been well studied, which have also been successfully solved by gradient-based solvers. We could, of course, enlarge our advantage by using problems where gradient-based methods fail. For instance, recently we applied our algorithm to battery electrode optimization (see figure below, to be published separately). There is no gradient-based method for such transient and non-linear problems to the best of our knowledge. Still, in this manuscript, we would like to use the problems that gradient-based methods can easily handle to verify the effectiveness and robustness of our approach. Thus, we did not make too much effort to show performance over gradient-based methods, but over gradient-free methods.

Reviewer #3 (Remarks to the Author):

1. Some related works are missing, i.e., 3D Topology Optimization using Convolutional Neural Networks; Deep Generative Design: Integration of Topology Optimization and Generative Models, etc. The authors shall spend more effort in justifying the novelty of the method.

Response:

We thank the reviewer for the suggestion. However, we would like to note that we already included the two papers the reviewer mentioned (Ref. 20 and 21 in the current version, 15 and 16 in the previous version). We discussed the differences of our approach and these generative methods (Introduction, Line 47 of current version):

“There are some reports on leveraging machine learning to reduce the computational cost of topology optimization^{19–25}. Generative models are used to predict solutions of the same problem under different conditions, after being trained by optimized solutions from gradient-based methods. For example, Yu, et al.²⁶ used 100,000 optimal solutions to a simple compliance problem with various boundary forces and the optimal mass fractions to train a neural network consisting of Convolutional Neural Network (CNN) and conditional Generative Adversarial Network (cGAN), which can predict near-optimal designs of mass fraction for any given boundary forces. However, these schemes are not topology optimization algorithms: they rely on existing optimal designs as the training data. The predictions are restricted by the coverage of the training datasets. To consider different domain geometry or constraints, new datasets and networks would be required. Besides, the designs predicted by the networks are close to, but still different from the optimal designs.”

Following the reviewer’s suggestion, we did more literature search and found a related topic that we did not cover, sequential model-based optimization. We add a new paragraph to discuss:

(Line 38) **Machine learning has been used in sequential model-based optimization (SMBO) targeting at expensive objective function evaluation^{13,14}. For instance, Bayesian optimization (BO)¹⁵ uses a Gaussian prior to approximate the conditional probability distribution of an objective $p(y|x)$ where $y = F(x)$ is the objective and x is the design variable (vector); then the unknown regions can be estimated by the probability model. In Covariance Matrix Adaptation Evolution Strategy (CMA-ES)¹⁶, a multivariable Gaussian distribution is used to sample new queries. However, these methods are not designed for large-scale and high-dimensional problems. Despite some improvement to scale up these algorithms^{17,18}, none of them has been implemented in topology optimization to the best of our knowledge.**

2. The experiment on the two examples is not enough to account for the claim that it can solve the optimization problems for large systems; The systems demonstrated is not complicated enough.

Response:

Typically for an initial work of an algorithm, it is often tested on problems that have been well studied so that the performance can be compared. Thus, we chose benchmark problems that have been studied which are not very complicated. From these simple examples, we show that we can converge to the global optimum faster than other gradient-free algorithms, such as CMA-ES, BO

and directly applying GSA. We could, of course, enlarge our advantage by using problems where gradient-based methods fail. For instance, recently we applied our algorithm to battery electrode optimization (see figure below, to be published separately) which was only showcasing rather than testing the algorithm. There is no gradient-based method for such transient and non-linear problems to the best of our knowledge.

In the current version, following the reviewer’s suggestion, we add two binary examples which are relatively more complicated. We show that our approach outperforms the gradient-based method and a recent algorithm based on reinforcement learning.

3. *Model complexity analysis is expected instead of just empirical running time comparison.*

Response:

We are not exactly sure about what “model complexity” specifically refers to here, so we discuss several scenarios. If the reviewer means the complexity of the topology optimization, we are treating the four problems as searching in the space $[0,1]^{25}$, $[0,1]^{121}$, $\{0,1\}^{160}$ and $\{0,1\}^{640}$, respectively. If the reviewer means how we reduce the complexity of topology optimization to a DNN optimization problem, it is hard to tell since we do not know the complexity of topology optimization (the binary fluid-structure optimization problems). We put effort to theoretically prove the convergence, and obtain a theoretical convergence rate regardless of the topology optimization problem:

$$(F(\hat{\rho}) - F^*)^2 \leq \tilde{O}\left(\frac{C}{\sqrt{n_{train}}}\right),$$

where n_{train} is the total number of training data C is a constant related to some inherent properties of F and DNN, F^* is the global minimum of F , and \tilde{O} omits log terms. A detailed theorem is shown in Supplementary Section 2:

$$\left(F(\hat{\rho}^{(T)}) - F(\rho^*)\right)^2 \leq 4C(T) \left(\frac{96B^2}{\sqrt{mT}} \sqrt{d^2 D \log(1 + 8BB_W^D D \sqrt{mTd})} + 12B^2 \sqrt{\frac{2\log^2 \delta}{mT} + \frac{8}{mT}} + \epsilon \right).$$

4. *The writing and organization can be improved.*

Response:

Following the reviewer's suggestion, we comprehensively revised our manuscript (including about 50% new materials). We also created a completely new 17-page Supplementary Information.

Partial list of major changes (Please see the manuscript for all changes)

1. Theoretical convergence rate

We add a theoretical upper bound of our convergence rate and gives the proof. The simplified version is presented in our main text:

This algorithm can converge provably under some mild assumptions. Given the total number of training data n_{train} , for any trained DNN with small training error, we have

$$(F(\hat{\rho}) - F^*)^2 \leq \tilde{O}\left(\frac{C}{\sqrt{n_{train}}}\right),$$

where C is a constant related to some inherent properties of F and DNN, F^* is the global minimum of F , and \tilde{O} omits log terms. This result states that when our trained DNN can fit the training data well, our algorithm can converge to the global optimal value.

The detailed theory and its derivation are elaborated in Supplementary Section 2.

2. More comparisons with state-of-the-art methods

In our existing two compliance minimization problems, we compare our approach with several gradient-free methods.

Fig. 1: Setup and results of a compliance minimization problem with 5×5 design variables. **a**, Problem setup. **b**, Best dimensionless energy with a total of n_{train} accumulated training samples. SOLO denotes our proposed method where the cross “X” denotes the convergence point

(presented in **e**), “Offline” denotes training a DNN offline and then uses GSA to search for the optimum without updating the DNN, SS denotes Stochastic Search, which is the same as SOLO except that $\hat{\rho}$ in each loop is obtained by the minimum of existing samples, CMA-ES denotes Covariance Matrix Adaptation Evolution Strategy, BO denotes Bayesian Optimization. SOLO converges the fastest among these methods. **c**, Energy prediction error of $\hat{\rho}$ relative to FEM calculation of the same material distribution. **d**, Optimized design produced by the gradient-based method. $\tilde{E} = 0.293$. **e**, Optimized design produced by SOLO. $n_{train} = 501$ and $\tilde{E} = 0.298$. **f**, Optimized design produced by SOLO. $n_{train} = 5,782$ and $\tilde{E} = 0.293$. In **d-f**, dark red denotes $\rho = 1$ and dark blue denotes $\rho = 0$, as indicated by the right color scale bar.

3. Two additional examples

We leveraged our algorithm to address discrete fluid-structure optimization. We show that our method outperforms the gradient-based method and a recent algorithm based on reinforcement learning.

Fig. 2: Setup and results of a fluid-structure optimization problem with 20×8 design variables. **a**, Problem setup. The vertical green line denotes the inlet while the vertical blue line denotes the outlet. **b**, Dimensionless inlet pressure versus n_{train} , the number of accumulated training samples. SOLO-G denotes a greedy version of our proposed method, SOLO-R denotes the regular version of our proposed method. The horizontal dashed line denotes the solution from the gradient-based method. The cross “X” denotes the convergence point (presented in **d** and **e**, respectively). **c**, Optimized design obtained by the gradient-based method. $\tilde{P} = 0.9569$. **d**, Optimized design obtained by SOLO-G. $n_{train} = 286$ and $\tilde{P} = 0.9567$. **e**, Optimized design obtained by SOLO-R. $n_{train} = 2,148$ and $\tilde{P} = 0.9567$. In **c-e**, black denotes $\rho = 1$ (solid) and white denotes $\rho = 0$ (void). These solutions are equivalent since the flow is blocked by the black squares forming the ramp surface and the white squares within the ramp at the left bottom corner are irrelevant.

Fig. 3: Setup and results of a fluid-structure optimization problem with 40×16 design variables. **a**, Problem setup. **b**, Dimensionless inlet pressure versus n_{train} , the number of accumulated training samples. SOLO-G denotes a greedy version of our proposed method, where the cross “X” denotes the convergence point (presented in **d**). The horizontal dashed line denotes the solution from the gradient-based method. **c**, Optimized design obtained by the gradient-based method. $\tilde{P} = 0.8065$. **d**, Optimized design obtained by SOLO-G. $n_{train} = 1,912$ and $\tilde{P} = 0.8062$. In **c,d**, black denotes $\rho = 1$ (solid) and white denotes $\rho = 0$ (void). The SOLO-G result in **d** has two gaps at the 7th and 12th columns, while the gradient-based result in **c** gives a smooth ramp. We try filling the gaps and find that their existence indeed reduces pressure, which demonstrates the powerfulness of our gradient-free method.

4. Others

There are additional minor changes in the main text. We also add more figures and a theory section in Supplementary Information.

REVIEWER COMMENTS

Reviewer #1 (Remarks to the Author):

Some of the concerns have been addressed in the revised manuscript. But I still have a few concerns.

1. It is quite confusing to use online and offline. It is an adaptive system that utilizes DNN to estimate the pressure and FEM to generate the energy and pressure of material distribution. A small neural network as shown in Fig. 6 is used for the adaptive training. Therefore, the system has to be retrained for every topology optimization. It is fair if same neural network is used in the offline training. A small neural network may not be sufficient to solve complicated problems due to under fitting.
2. What is gradient-free? As stated in the paper, "For the fluid problems, the gradient-based method produces a continuous array, and we use multiple thresholds to convert it to binary arrays and recompute their objective (pressure) to select the best binary array", does it mean the system still uses gradient? If the gradient-free method refers to the optimizer in DNN, I think it is not a contribution of the paper. Because ADAM optimizer has been widely used.
3. The samples used in the training are not clearly explained.
4. In Fig. 4b and Fig. 5b, the improvement with SOLO is only 0.02%, which could be smaller than the fluctuation value in the training.

Reviewer #2 (Remarks to the Author):

The authors have considerably improved the manuscript regarding the questions that were raised by the reviewers.

From my perspective, many issues have now been resolved, especially with respect to unclear statements in the original version.

On the other hand, some issues are still open and probably need further elaboration.

Firstly, I feel unsure about the application studies that have been performed. The authors provide only simple examples, which neither let one allow to judge scalability nor applicability to real use cases. Since I'm not from the community, I may not know how the "rules" in this community are regarding application examples. Personally I would have wished for a more practical demonstration.

Secondly, several times it is said that the method is gradient-free. Is it really? Or do you also rely on gradients during the optimization process?

Further, it is said that "For gradient-free methods, there is no similar concept like "residuals". How does this compare to the convergence evaluation, where you show that the residual is bound?

Thirdly, I found it very difficult to follow the derivation of the convergence proof, and to understand the implication of made assumptions and parameters in the derived bound. In particular, it seem that a global optimum is assumed here. Where does this come from? Isn't it even questionable that a global optimum exist since it depends on grid resolution etc.

Reviewer #3 (Remarks to the Author):

I appreciate the authors' effort in revising the paper. They addressed some of my previous concerns.

However, I think the novelty of the proposed method is quite limited for this journal. Some important references are not discussed and compared. Some more sophisticated methods such as convolutional

neural networks[a], deep belief networks (DBNs)[c], Deep Reinforcement Learning[b] are already investigated in the literature. It is not clear why the simpler DNN should be better, what is the intuition behind it?

To me, the self-directed learning process seems to be quite straightforward in the deep learning field.

The evaluation can be performed better. The examples shown in the paper do not look interesting. I expect some real-world problems with larger and noisier data to be studied as well.

The writing and organization of the paper make it much harder to understand.

minor comments:

I do not think figure 6 is needed.

references

(a) 3D Topology Optimization Using Convolutional Neural Networks

(b) Deep Reinforcement Learning-Based Topology Optimization for Self-Organized Wireless Sensor Networks

(c) Topology Optimization Accelerated by Deep Learning

Response to the Reviewer's Comments

Dear editor and reviewers:

Thank you for your careful reviews and suggestions to our manuscript. We have studied the comments and made major changes in the manuscript. We have addressed all comments in the revised manuscript. All revised/new parts were highlighted in red the manuscript. Following the reviewers' suggestions, we comprehensively revised our manuscript. We have also added calculations and analysis on two new classes of problems.

Reviewer #1 (Remarks to the Author):

1. It is quite confusing to use online and offline. It is an adaptive system that utilizes DNN to estimate the pressure and FEM to generate the energy and pressure of material distribution. A small neural network as shown in Fig. 6 is used for the adaptive training. Therefore, the system has to be retrained for every topology optimization. It is fair if same neural network is used in the offline training. A small neural network may not be sufficient to solve complicated problems due to under fitting.

Response:

We agree with the reviewer that the network has to be retrained for every topology optimization. However, this does not conflict with the definition of “online”. In the machine learning field, online learning can be defined as “a method of machine learning in which data becomes available in a sequential order and is used to update the best predictor for future data at each step, as opposed to batch learning techniques which generate the best predictor by learning on the entire training data set at once” [a]. In the optimization field, there is a description saying “... In the former (online methods), CNN is trained during an optimization process, whereas CNN is trained in the learning phase prior to the optimization in the latter case (offline methods)” [b]. The meaning of “online” and “offline” in our manuscript accords with both descriptions. We would like to mention that retraining is also needed using other methods for new topology optimization (e.g., different domain geometry or constraints), as noted in the introduction of the manuscript.

For fair comparison, we used the same neural network when comparing between online and offline methods. They have the same architecture, learning rate, batch size, etc. The only difference is how training data is generated. For the online methods, training data is generated dynamically according to the prediction of the DNN, whereas for the offline methods, training data is generated prior to training. As a result, the parameters (weights) of the networks are different and so are the optimized solutions from the networks.

We agree with the reviewer that underfitting could become an issue when the problem becomes larger and more complicated. Our strategy was to use a relatively large network (to avoid underfitting) with regularization (to avoid overfitting). Although the network in Fig. 6 (Fig. 8a in this revised manuscript) looks small, it is sufficient to approximate the objective functions, thanks

to the powerful capabilities of DNN. If the problem becomes more complicated, a larger network can be used. For instance, in our newly added problems, we used larger neural networks, as shown in Fig. 8 (attached below). Specially, in the heat transfer problem, our DNN grows during optimization.

Fig. 8: Architectures of DNN. The input is a design vector ρ and the output is the predicted objective function value $f(\rho)$. “Linear” presents a linear transformation and “BatchNorm1d” denotes one-dimensional batch normalization used to avoid internal covariate shift and gradient explosion for stable training⁵⁵. “LeakyReLU” is an activation function extended from ReLU with activated negative values. “Dropout” is a regularization method to prevent overfitting by randomly masking nodes⁵⁹. **a**, the DNN in the compliance and fluid problems. **b**, the DNN in the heat problem. Two architectures are used in this problem. At the 100th loop and before, $B = 1$, $C = 512$, and the Linear layer in the dashed box is 512×256 . At the 101st loop and afterwards, $B = 4$, $C = 512$ and the 4 Linear layers are 256×512 , 512×512 , 512×512 and 512×256 , respectively. **c**, the DNN in the truss optimization problems. $B = 1$. $C = 512$ when $N = 72/432$; $C = 1,024$ when $N = 1,008$.

[a] https://en.wikipedia.org/wiki/Online_machine_learning

[b] Sasaki, Hidenori, and Hajime Igarashi. "Topology optimization accelerated by deep learning." IEEE Transactions on Magnetics 55.6 (2019): 1-5.

2. What is gradient-free? As stated in the paper, “For the fluid problems, the gradient-based method produces a continuous array, and we use multiple thresholds to convert it to binary arrays and recompute their objective (pressure) to select the best binary array”, does it mean the system still uses gradient? If the gradient-free method refers to the optimizer in DNN, I think it is not a contribution of the paper. Because ADAM optimizer has been widely used.

Response:

“Gradient-free” means that we do not need to evaluate the gradient of the objective function F during optimization (This term does not prevent using internal gradients by DNN itself, e.g., the learning algorithm of DNN may use gradient matrices with respect to weights for back propagation, but this is internal of DNN and has nothing to do with the optimization objective function F). The meaning of “Gradient-free” here is consistent with the typical use of this term such as in “gradient-based algorithms” or “gradient-free methods”. For instance, Bayesian Optimization is commonly regarded as a gradient-free method although it uses gradient to calculate the minimum of the surrogate model; Ref. [a] regarding Bayesian Optimization writes “When we evaluate f , we observe only $f(x)$ and no first- or second-order derivatives. This prevents the application of first- and second-order methods like gradient descent, Newton’s method, or quasi-Newton methods. We refer to problems with this property as ‘derivative-free’.”

In the sentence “For the fluid problems, the gradient-based method produces a continuous array...”, the “gradient-based method” in the sentence does not refer to our method. It refers to the gradient-based method that we used as a baseline to compare with our gradient-free method. To avoid confusion, we changed this sentence to “For the fluid problems, the gradient-based **baseline** method produces a continuous array...”

Here “gradient-free method” does not refer to the optimizer in DNN. For the DNN, it uses a gradient-based optimizer ADAM to find the minimum of loss function. Training DNN with ADAM does not conflict with the term “gradient-free” since this is an internal part of DNN training, which has nothing to do with the gradient of the optimization objective function. We thank the reviewer for the question. Following the reviewer’s comment, we add a sentence to better explain “gradient-free”: (Line 29)

“**Gradient-free optimizers, i.e. without calculating the gradient or derivative of the objective function, have been attempted by several researchers, most of which are stochastic and heuristic methods.**”

[a] Frazier, Peter I. "A tutorial on Bayesian optimization." arXiv preprint arXiv:1807.02811 (2018).

3. The samples used in the training are not clearly explained.

Response:

Generally speaking, we use the pairs of $\boldsymbol{\rho} - F(\boldsymbol{\rho})$ to train the network. As for how the pairs (especially $\boldsymbol{\rho}$) are generated, it slightly depends on problems (discussed in the next paragraph). Following the reviewer’s comments, we revised and expanded the descriptions in several places.

(Line 113) “**A small batch of random vectors (or arrays) $\boldsymbol{\rho}$ satisfying the constraints in Eq. (2) is generated. The corresponding objective values $F(\boldsymbol{\rho})$ are calculated by FEM. Then, $\boldsymbol{\rho}$ and $F(\boldsymbol{\rho})$** ”

are inputted into the DNN as the training data so that the DNN has a certain level of ability to predict the function values based on the design variables. Namely, the output of the DNN $f(\boldsymbol{\rho})$ approximates the objective function $(\boldsymbol{\rho})$.”

For the two compliance minimization problems, we wrote (Line 162) “we use 100 random arrays to initialize the DNN. Then Generalized Simulated Annealing (GSA) is used to obtain the minimum $\hat{\boldsymbol{\rho}}$ based on the DNN's prediction. Afterwards, 100 additional samples will be generated by adding disturbance to $\hat{\boldsymbol{\rho}}$.” The details of disturbance are expressed in the Methods-Random generation of new samples from a base design (Line 416). For the fluid-structure optimization problems, we used two variants. (Line 225) “One is denoted as SOLO-G, a greedy version of SOLO where additional 10 samples produced in each loop are all from the DNN's prediction. The initial batch is composed of a solution filled with zeros and 160 solutions each of which has a single element equal to one and others equal to zero. The pressure values corresponding to these designs are calculated by FEM. These 161 samples are used to train a DNN. Next, Binary Bat Algorithm (BBA) is used to find the minimum of the DNN. The top 10 solutions (after removing repeated ones) encountered during BBA searching will be used as the next batch of training data. The other variant, denoted as SOLO-R, is a regular version of SOLO where each loop has 100 incremental samples. 10 of them are produced in the same way as SOLO-G whereas the rest 90 are generated by adding disturbance to the best solution predicted by the DNN. Similar to the compliance minimization problems, the disturbance includes mutation and crossover.” For the new heat transfer problem, (Line 291) “Our method SOLO is initialized by 500 random samples to train a DNN. Bat Algorithm (BA) is then used to find the minimum of the DNN, based on which additional 200 samples are generated in each loop.” For the new truss optimization problems, (Line 327) “SOLO is initialized by 100, 500 and 1000 samples, respectively. The number of incremental samples per loop is 10% of the initialization samples. 10% of incremental samples are the optima obtained by BA based on the DNN's prediction, and the rest 90% are generated by mutation, crossover and convolution of the best solution predicted by the DNN.”

We would be happy to further adding more detail if the reviewer has questions on any specific places.

4. In Fig. 4b and Fig. 5b, the improvement with SOLO is only 0.02%, which could be smaller than the fluctuation value in the training.

Response:

These two fluid-structure optimization problems can be well addressed by gradient-based methods; thus, we choose these problems as benchmarks so that we know the optima. Accordingly, there is not much room for improvement. Our goal is not to exceed the gradient-based baseline by a large margin in these problems. We want to show that our algorithm can give excellent solutions which are equal to or even better than gradient-based baselines. In more complicated problems (such as

those we added in this revised manuscript, especially truss optimization) which gradient-based methods cannot address, our method would be more suitable.

As for “fluctuation”, if the reviewer means that the DNN may not be able to resolve 0.02% difference, we have used a mechanism to alleviate this issue. The resolution of the DNN can automatically increase because of normalization. As we wrote in the manuscript, (Line 407) “all inputs are normalized before training”, $F(\boldsymbol{\rho})$ values will be normalized to have zero mean and unit deviation. As a result, with more loops, there will be more samples close to the current optimum and thus decrease the standard deviation of raw data. After normalization, the resolution near the optimum becomes higher.

If the reviewer means that the fluctuation causes randomness in the results, we have added test of the robustness by repeating the experiments. In fact, training fluctuation is not the only source of randomness. Generation of new samples and finding the optimum of $f(\boldsymbol{\rho})$ will also contribute to randomness. In the manuscript, we add (Line 244) “To account for randomness, we repeat the experiments another four times and the results are similar to Fig. 4b (see Supplementary Figs. 5 for and 6)” and (Line 251) “Similar trends can be observed when repeating the experiments (see Supplementary Fig. 7)”. The supplementary figures (attached below) show four new additional experiments for each algorithm. We can see that the conclusion remains the same.

Supplementary Fig. 5: Repeating SOLO-G for the fluid-structure optimization problem with 20×8 mesh. All configurations are the same as Fig. 4b except different random seeds. They obtain the same objective \tilde{P} despite different convergence rate.

Supplementary Fig. 6: Repeating SOLO-R for the fluid-structure optimization problem with 20×8 mesh. All configurations are the same as Fig. 4b except different random seeds. They obtain the same objective \tilde{P} despite different convergence rate.

Supplementary Fig. 9: Repeating SOLO-G for the fluid-structure optimization problem with 40×16 mesh. All configurations are the same as Fig. 5b except that different random seeds and higher n_{train} are used. They all outperform the gradient-based baseline.

Reviewer #2 (Remarks to the Author):

1. I feel unsure about the application studies that have been performed. The authors provide only simple examples, which neither let one allow to judge scalability nor applicability to real use cases. Since I'm not from the community, I may not know how the "rules" in this community are regarding application examples. Personally I would have wished for a more practical demonstration.

Response:

We thank the reviewer for the feedback. Following the comments, we added solving two new types of problems as examples to demonstrate applicability and scalability. These added examples made our manuscript more comprehensive, so we really appreciate the reviewer's suggestion. These new results added several new pages. To be concise, we summarize the results below (instead of copying all pages) and point to the pages.

Fig. 6: Setup and results of a heat transfer enhancement problem with 10×10 design variables. **a**, Engineering background: a group of copper pipes are inserted in a phase change material. Because of symmetry, we only need to consider 1/8 of the unit cell (dark blue area in the top right corner). **b**, Problem setup. The black dots denote locations of design variables. **c**, Dimensionless charging time versus n_{train} , the number of accumulated training samples. SOLO denotes our proposed method, where the cross “X” denotes the convergence point (presented in **d**). “Direct” denotes solving the problem directly by gradient descent. “Approximated” denotes simplifying this problem to a steady state problem. **d**, Optimized design obtained by SOLO. $\tilde{t} = 0.0137$. **e**, Optimized design obtained by “Direct”. $n_{train} = 24,644$ and $\tilde{t} = 0.0275$. **f**, Optimized design obtained by “Approximated”. $\tilde{t} = 0.0203$. In **d-f**, black denotes $\rho = 1$ (copper) and white denotes $\rho = 0$ (wax). The SOLO result in **d** has islands isolated from major branches, while the “Approximated” result in **f** gives a connected structure. We try combining the islands to be part of major branches and find that the existence of isolated islands indeed reduces time, which demonstrates the powerfulness of our gradient-free method.

We added a heat transfer enhancement problem, which is time-consuming to compute, to investigate enhancing heat conduction in a phase change material to demonstrate applicability. The main result figure is attached above. The details are in Pages 10-13. We looked into a complicated, transient and non-linear problem where the boundary condition is changing to replicate the applications in buildings or battery packages. We showed a superior performance compared with a traditional method (called “Approximated” in our manuscript) which approximates the problem by a constant boundary condition. Also note that the tradition method has to use approximation to be able to solve this problem, which highlights the capability of our method. This example demonstrates the application to solving complicated engineering problems.

Fig. 7: Setup and results of three truss optimization problems with different numbers of bars (equal to the numbers of design variables). a, Illustration of an antenna tower, an exemplary application of truss structures. b, Illustration of the problem setup. The block is repeated until the given number of bars is reached. c-e, Dimensionless weight \tilde{W} versus the number of accumulated training samples n_{train} . SOLO denotes our proposed method. BA denotes Bat Algorithm. The numbers of bars for these three sub-figures are 72, 432 and 1,008, respectively. Each experiment is repeated five times; the curves denote the mean and the shadows denote the standard deviation.

Also, we added three truss optimization problems to test the scalability by using up to 1,008 design variables and a design space of $10^{1,214}$ possible solutions. The result figure is attached above. Details are shown in Pages 13-15. We compare with a heuristic method Bat Algorithm (BA) to show calculation reduction of over three orders of magnitude. This example demonstrates the application to scaling up and solving large size engineering problems with many design variables.

2. Several times it is said that the method is gradient-free. Is it really? Or do you also rely on gradients during the optimization process? Further, it is said that "For gradient-free methods, there is no similar concept like "residuals". How does this compare to the convergence evaluation, where you show that the residual is bound?

Response:

Here "gradient-free" means that we do not need to evaluate the gradient of the objective function F during optimization (This term does not prevent using internal gradients by DNN itself, e.g., the learning algorithm of DNN may use gradient matrices with respect to weights for back propagation, but this is internal of DNN and has nothing to do with the optimization objective function F). The meaning of "Gradient-free" here is consistent with the typical use of this term such as in "gradient-based algorithms" or "gradient-free methods". For instance, Bayesian Optimization is commonly regarded as a gradient-free method although it uses gradient to calculate the minimum of the surrogate model; Ref. [a] regarding Bayesian Optimization writes "When we evaluate f , we observe only $f(x)$ and no first- or second-order derivatives. This prevents the application of first- and second-order methods like gradient descent, Newton's method, or quasi-Newton methods. We refer to problems with this property as 'derivative-free'." Following the reviewer's comment, we add a sentence to better explain "gradient-free": (Line 29)

"Gradient-free optimizers, i.e. without calculating the gradient or derivative of the objective function, have been attempted by several researchers, most of which are stochastic and heuristic methods."

During optimization, we do not rely on the gradients of the objective function. Although gradient matrices with respect to weights are used internally by the DNN learning algorithm, it does not conflict with the term "gradient-free" since it has nothing to do with the gradient of the optimization objective function.

In the previous manuscript and this revised manuscript, we did not use "residuals" anywhere. The term "residuals" is used by the previous reviewer who asked to show the residuals for gradient-based baseline and our method for a fair comparison. So "residuals" was only used in the response letter. To follow with the reviewers' wording, here we still borrow the term "residuals", only for the discussion below. First we list some meanings of "residual" to avoid confusion.

1. In an iterative linear solver which, for instance, solves $\mathbf{Ax}=\mathbf{b}$, the difference between left-hand side and right-hand side can be called "residual". We use "residual $\textcircled{1}$ " to denote this meaning.
2. (Extension of "residual $\textcircled{1}$ ") In gradient-based methods, a typical way to evaluate convergence is to observe gradient $\nabla F(\boldsymbol{\rho})$, since we need $\nabla F(\boldsymbol{\rho}) = \mathbf{0}$ at the optimum. This gradient $\nabla F(\boldsymbol{\rho})$ can also be used to define residual, for instance, by $\|\nabla F(\boldsymbol{\rho})\|$. This meaning is also known as optimality tolerance or first-order optimality measure. We will denote it as "residual $\textcircled{2}$ ".

3. During an optimization process, the optimized solution $F(\hat{\rho}^{(n)})$ will change with the number of iterations n . We can use $|F(\hat{\rho}^{(n+1)}) - F(\hat{\rho}^{(n)})|$, i.e. the change in the value of the objective function, to define “residual”. We will denote it as “residual③”.
4. (Extension of “residual③”) Instead of using adjacent two steps $n + 1$ and n , we can use $\left| \min_{i \leq n+m} F(\hat{\rho}^{(i)}) - \min_{i \leq n} F(\hat{\rho}^{(i)}) \right|$, i.e. the change of the best solution from the n -th step to the $(n + m)$ -th step. This definition is useful when $F(\hat{\rho}^{(n)})$ is stochastic (as n increases, it decreases with fluctuation). We will denote it as “residual④”.
5. The difference between the optimized solution and the true optimum, namely, $|F(\hat{\rho}) - F^*|$. We denote it as “residual⑤”.

In the previous response where we said “for gradient-free methods, there is no similar concept like ‘residuals’ ”, we understood the term as ①, ② or ③”. Since our method does not solve any equation like $\nabla F(\rho) = 0$, does not monitor the gradients, and does not calculate differences between adjacent two steps, we said there is no similar concept in our method. Residual④ is a good indicator of the degree of convergence. In our proposed method SOLO, we used this metric to qualitatively determine whether the algorithm converged: we observed the solution $F(\hat{\rho}^{(n)})$ and stopped the algorithm when $\min_{i \leq n} F(\hat{\rho}^{(i)})$ did not decrease much with n (although we did not enforce a strict criteria). In principle, residual④ can be used by both gradient-based baselines and our proposed method SOLO to give a “fair” comparison (we do not feel this is fair since the computation time is quite different). Yet in our problems, gradient-based methods are treated as baselines. We want the real optimum from them and thus set a more restricted convergence criterion. In another word, if we make the comparison “fair” by setting the same residual④, we will see more loops and probably better results in SOLO.

One cannot use “residual⑤” to check for convergence in numerical calculations since we do not know the true optimum. We proved that “residual⑤” is bounded, which shows that the method is guaranteed to converge. The bound gives an estimation of the difference between the optimized solution and the true optimum. As is typical for theoretic bound estimation, the theoretic bound is not problem-specific and loose. Using a theoretic bound will always overestimate the needed number of iterations, and because the bound is loose, this overestimation is often very large. In practice, a problem always converges much faster. For practical applications, one often judges convergence by the relative change of an indicator between iterations. When the change is small, we say it is converged. Here we use “residual④” as our convergence indicator.

[a] Frazier, Peter I. "A tutorial on Bayesian optimization." arXiv preprint arXiv:1807.02811 (2018).

3. I found it very difficult to follow the derivation of the convergence proof, and to understand the implication of made assumptions and parameters in the derived bound. In particular, it seem that a global optimum is assumed here. Where does this come from? Isn't it even questionable that a global optimum exist since it depends on grid resolution etc.

Response:

Following the reviewer's comment, we added a proof sketch. More explanations are also added. Please see Supplementary Section 2.3 (Page 16 in Supplementary Information).

In our manuscript, we wrote that our algorithm was applied to the following problem (Eq. (2))

$$\begin{cases} \min_{\boldsymbol{\rho}=(\rho_1,\rho_2,\dots,\rho_N)} F(\boldsymbol{\rho}) \\ G_0(\boldsymbol{\rho}) = \sum_{i=1}^N w_i \rho_i - V_0 \leq 0 \\ G_j(\boldsymbol{\rho}) \leq 0, \quad j = 1, \dots, M \\ \rho_i \in S, \quad i = 1, \dots, N \end{cases}$$

This means that our algorithm gets involved after discretizing the domain into finite elements. So a global optimum is assumed on a given grid (i.e. a finite system). In fact, for practical applications, the global optimum assumption is reasonable since $F(\boldsymbol{\rho})$ has a physical meaning and it must have a minimum value. Mathematically, it is possible to construct a function without a minimum value and goes to $-\infty$, but we do not have such situations in practical applications. The global optimum is reached when $F(\boldsymbol{\rho})$ reaches its minimum value. Note that convergence is measured by $F(\boldsymbol{\rho})$, so convergence to global optimum allows the existence of multiple design variable solutions as long as they give the minimum $F(\boldsymbol{\rho})$. Also, there are multiple ways to discretize a domain and prevent grid-dependent solutions (this is a genetic topic for all methods and there are some common techniques), though this is a separate topic outside the scope of this manuscript.

Reviewer #3 (Remarks to the Author):

1. I think the novelty of the proposed method is quite limited for this journal. Some important references are not discussed and compared. Some more sophisticated methods such as convolutional neural networks[a], deep belief networks (DBNs)[c], Deep Reinforcement Learning[b] are already investigated in the literature. It is not clear why the simpler DNN should be better, what is the intuition behind it? To me, the self-directed learning process seems to be quite straightforward in the deep learning field.

References

(a) 3D Topology Optimization Using Convolutional Neural Networks

(b) Deep Reinforcement Learning-Based Topology Optimization for Self-Organized Wireless Sensor Networks

(c) Topology Optimization Accelerated by Deep Learning

Response:

We thank the reviewer for the suggestion. We expanded our reference and discussions following the comments. We would like to note that some papers use more advanced and complex network architectures than our simple fully connected layers. However, the key is not network architecture; our novelty is that we use a completely different paradigm from them. As we mentioned in our manuscript, most of them are *generative models*, which accelerate topology optimization by training the network by optimal solutions which are calculated *a priori* (by gradient-based solvers). By contrast, we look at complicated problems where gradient-based solvers fail; there do not exist prior optimal solutions as training data and we use design-objective pairs to train the network.

We avoid to vaguely say we are “better” than them, because we are in different scopes and it is not an apple-to-apple comparison. More accurately, we claim our method makes it possible to solve highly complicated problems (which gradient-based solvers cannot deal with) in short time; in this scenario, the generative methods in literature cannot be applied.

For the three references, we first discuss them one-by-one. We have added reference/comparison to them which will be discussed next.

(a) 3D Topology Optimization Using Convolutional Neural Networks. This paper does not use *generative models*, but their approach is very similar to and less applicable than *generative models*. This paper and *generative models* both use optimal solutions to train the networks, and output predicted optimal solutions, yet the former inputs intermediate solutions (during gradient-based optimization) which leads to a restricted application scenario and the latter only need boundary conditions, problem settings, etc. Therefore, we focus on discussing *generative models* in our Introduction.

(b) Deep Reinforcement Learning-Based Topology Optimization for Self-Organized Wireless Sensor Networks. We do not think this paper is relevant to our topic after reading it. Although this paper works on “topology optimization”, this term in communication networks has a different

meaning. The term “topology optimization” means the connection of nodes in communication networks. By contrast, this term means material distribution in the field of our manuscript. The difficulty to address in our fields, extremely high dimension, is not an issue in the communication field. Therefore, we choose not to discuss this paper in our manuscript. We noticed that this paper uses deep reinforcement learning, and we did include another relevant paper based on reinforcement learning to compare with: (Line 238) “We obtain the optimum better than the gradient-based method after only 286 FEM calculations. For comparison, a recent topology optimization work based on reinforcement learning used the same geometry setup and obtained the same solution as the gradient-based method after thousands of iterations³³; our approach demonstrates better performance.”

(c) Topology Optimization Accelerated by Deep Learning. This paper uses an offline scheme. We compared our proposed online method against the offline method in the two compliance minimization problems, as shown in Figs. 2 and 3. In the following, we first discuss the relation between our offline method with theirs and compare our proposed online method with theirs. As a brief summary, our method is quite different from theirs. Our method is, from some perspectives, much faster.

1) Offline methods. Their method and our offline method both train the network prior to optimization, freeze the network during optimization and predict objective function values, but there are major differences. In our optimization (not only our offline method, but also all algorithms), we assume no prior knowledge of the solutions, thus the network is trained on purely random data, and the network gives an optimized solution with no more FEM calculations. In contrast, they first calculate the solutions to problems with objectives or configurations similar to the problems they are looking at (in the following, we call the former “auxiliary problems” and latter “main problems” for simplicity). To solve the main problems, they train a network by the data sampled from the designs searched by genetic algorithm (GA) when solving auxiliary problems. During optimization phase, searching is controlled by GA, and objectives are mostly calculated by the network and partially calculated by FEM: if the network gives a good design (whether a design is good or not is based on the knowledge of auxiliary problems) they perform FEM calculations to confirm the objective of the design. In summary, compared with our offline method, the advantage of their method is that it is more sophisticated and successfully reduces the number of FEM calculations to 10%~36%, the disadvantage is that it requires auxiliary problems.

2) Comparison between our online method and their offline method. Considering the auxiliary problems, it is hard to apply their method to our problems in the manuscript, since our problems are very expensive to directly obtain solutions from a heuristic method. Also, it is not fair when our methods do not use auxiliary problems. Still, we can present a comparison here, although it is not accurate since the problems are different. Even not considering the overhead computation of auxiliary problems (negative improvement if considering the auxiliary problems), their method reduces the number of FEM calculations to 10%~36%, yet our proposed online method reduces the number of FEM calculations by

2~5 orders of magnitude (i.e., reduced to 0.01%~1%). Therefore, ours greatly outperforms their method.

3) We referred this paper by saying (Line 64) “An offline learning method³¹ replaces some FEM calculations during the optimization process with DNN’s prediction, yet gives limited improvement especially considering it requires the solutions to similar problems for training.”

To give a more comprehensive review of literature, we added more references including the two suggested by the reviewer, and more discussions on similar methods:

(Line 39) “As a trade-off, sometimes searching space can be reduced in order for less computation. For instance, pattern search has been applied^{13,14} which is a non-heuristic⁴¹ method with a smaller searching space but is more likely to be trapped in local minima.”

(Line 48) “... However, as demonstrated later in the paper, these methods are not designed for large-scale and high-dimensional problems, and thus do not perform well in topology optimization for slow convergence¹⁹ or requirement of shrinking design space²⁰. Despite some improvement to scale up these algorithms^{21,22}, none of them has shown superior performance in topology optimization to the best of our knowledge.”

2. The evaluation can be performed better. The examples shown in the paper do not look interesting. I expect some real-world problems with larger and noisier data to be studied as well.

Response:

Following the reviewer’s comments, we added two new types of real-world problems to evaluate the performance. We used a heat transfer enhancement problem (which is complicated and time-consuming to compute) with noise data to demonstrate applicability to complex problems and truss optimization problems with large data to demonstrate scalability. These added examples made our manuscript more comprehensive, so we really appreciate the reviewer’s suggestion. These new results added several new pages. To be concise, we summarize the results below (instead of copying all pages) and point to the pages.

We added a heat transfer enhancement problem, which is time-consuming to compute, to investigate enhancing heat conduction in a phase change material to demonstrate applicability. The main result figure is attached below. Details are in Pages 10-13. The data is noisier than previous examples since the time period we simulate and accordingly the time steps in the FEM solver both depend on the designs. Ideally a small change in the design lead to a small change in the objective function, yet due to limited accuracy of FEM in transient and boundary-varying problems (we used a lower FEM resolution for reasonable computation time), the change in the objective function becomes larger in practice. Our method is robust to such noise. We showed a superior performance compared with a traditional method (called “Approximated” in our manuscript) which approximates the problem by a constant boundary condition. Also note that the tradition method

has to use approximation to be able to solve this problem, which highlights the capability of our method. This example demonstrates the application to solving complicated engineering problems.

Fig. 6: Setup and results of a heat transfer enhancement problem with 10×10 design variables. **a**, Engineering background: a group of copper pipes are inserted in a phase change material. Because of symmetry, we only need to consider 1/8 of the unit cell (dark blue area in the top right corner). **b**, Problem setup. The black dots denote locations of design variables. **c**, Dimensionless charging time versus n_{train} , the number of accumulated training samples. SOLO denotes our proposed method, where the cross “X” denotes the convergence point (presented in **d**). “Direct” denotes solving the problem directly by gradient descent. “Approximated” denotes simplifying this problem to a steady state problem. **d**, Optimized design obtained by SOLO. $\tilde{t} = 0.0137$. **e**, Optimized design obtained by “Direct”. $n_{train} = 24,644$ and $\tilde{t} = 0.0275$. **f**, Optimized design obtained by “Approximated”. $\tilde{t} = 0.0203$. In **d-f**, black denotes $\rho = 1$ (copper) and white denotes $\rho = 0$ (wax). The SOLO result in **d** has islands isolated from major branches, while the “Approximated” result in **f** gives a connected structure. We try combining the islands to be part of major branches and find that the existence of isolated islands indeed reduces time, which demonstrates the powerfulness of our gradient-free method.

Also, we added three truss optimization problems to test the scalability by using up to 1,008 design variables and a design space of $10^{1,214}$ possible solutions. The result figure is attached below. Details are shown in Pages 13-15. The data amount reaches 5×10^4 for our method and 10^8 for the baseline. We compare with a heuristic method Bat Algorithm to show calculation reduction of over three orders of magnitude. This example demonstrates the application to scaling up and solving large size engineering problems with many design variables.

Fig. 7: Setup and results of three truss optimization problems with different numbers of bars (equal to the numbers of design variables). **a**, Illustration of an antenna tower, an exemplary application of truss structures. **b**, Illustration of the problem setup. The block is repeated until the given number of bars is reached. **c-e**, Dimensionless weight \tilde{W} versus the number of accumulated training samples n_{train} . SOLO denotes our proposed method. BA denotes Bat Algorithm. The numbers of bars for these three sub-figures are 72, 432 and 1,008, respectively. Each experiment is repeated five times; the curves denote the mean and the shadows denote the standard deviation.

3. The writing and organization of the paper make it much harder to understand.

Response:

We comprehensively revised the manuscript. Here we give a brief overview of our organization.

In Section Formulation, we proposed to solve the following problem

$$\min_{\rho=(\rho_1,\rho_2,\dots,\rho_N)} F(\rho)$$

$$\begin{cases} G_0(\rho) = \sum_{i=1}^N w_i \rho_i - V_0 \leq 0, \\ G_j(\rho) \leq 0, \quad j = 1, \dots, M \\ \rho_i \in S, \quad i = 1, \dots, N \end{cases}$$

where the domain of design variable ρ_i could be binary (e.g., 0/1), discrete (e.g., 0,1,2,3,...K) and continuous (e.g., from 0 to 1). Our idea is to use DNN as a surrogate model and dynamically generate training data to train the DNN. An intuitive explanation is shown in the figure below. The objective function F (black curve) is unknown to the algorithm. We would like to find the

minimum (black star) in an efficient way. We first randomly sample some data points (light blue dots) to initialize. The DNN f_1 (dashed light-blue line) trained on first batch of data only gives a rough representation of the true objective function F (solid black line). The second batch training data (dark blue dots) are generated by adding disturbance (orange curve) to the minimum of f_1 . After trained with two batches, the DNN f_2 (dashed dark-blue line) is more refined around the minimum (the region of interest), while remains almost the same at other locations such as the right convex part. f_2 is very close to finding the exact global minimum point.

This algorithm has a guaranteed convergence rate

$$(F(\hat{\rho}) - F^*)^2 \leq \tilde{O}\left(\frac{C}{\sqrt{n_{train}}}\right),$$

where $\hat{\rho}$ is the minimum found by our method, C is a constant related to some inherent properties of F and DNN, F^* is the global minimum of F , and \tilde{O} omits log terms.

In Section Results, we applied the algorithm to eight classic examples of four types (covering binary, discrete and continuous variables): two compliance minimization problems, two fluid-structure optimization problems, a heat transfer enhancement problem and three truss optimization problems.

We are happy to revise more if the reviewer has questions on any specific places.

4. *Minor comments: I do not think figure 6 is needed.*

Response:

Thanks for the suggestion. The previous Fig. 6 was indeed too simple and could be removed. In this version, we used more DNN architectures, so we redrew the figure with more plots (Fig. 8 in this version, also attached below).

REVIEWER COMMENTS

Reviewer #1 (Remarks to the Author):

All of my concerns have been addressed. I have no further comments.

Reviewer #2 (Remarks to the Author):

The current version of the manuscript has been improved again, including a number of new experiments and clarifications. However, there are remaining issues that need to be addressed in a better way: Firstly, I'm still confused about the term "gradient-free". It is said that no gradient wrt the objective function is used, yet internally the optimizer computes gradients to perform back propagation. It needs to be described to which loss (objective) function these gradients are computed. Secondly, the additional examples need further explanations. It seems that the same truss structure is repeated to build the antenna. What exactly is optimized for in this example? Also for the copper-wax example it is not clear to me what the objective function is. Is there a volume constraint? Is compliance what you aim to optimize for?

Finally, I was irritated about the statement "We agree with the reviewer that the network has to be retrained for every topology optimization" What does this mean? Please comment on the implications of this in real-world application where usually new designs want to be developed and tested.

Reviewer #3 (Remarks to the Author):

It is appreciated that the authors spend a lot of effort in revising the manuscript.

The authors have addressed most of my concerns (1) discussion on related works; (2) real-world examples; (3) improvements on the organization and writing.

I agree with other reviewers that the phrase "gradient-free" used in the paper is quite confusing. Although the authors added a sentence for clarification, I think it can still be misleading.

Overall, I still think that the contribution of the paper is limited. The results are incremental to the field, and the findings are not very exciting. So I believe the impact will be limited as well.

Response to the Reviewer's Comments

Dear editor and reviewers:

Thank you for your careful reviews and suggestions to our manuscript. We have studied the comments and made corresponding changes in the manuscript. We have addressed all comments in the revised manuscript. Following the reviewers' suggestions, we revised our manuscript and highlighted the changes in red.

Reviewer #1 (Remarks to the Author):

All of my concerns have been addressed. I have no further comments.

Response: We thank the reviewer for all the inputs and comments.

Reviewer #2:

The current version of the manuscript has been improved again, including a number of new experiments and clarifications. However, there are remaining issues that need to be addressed in a better way:

1. I'm still confused about the term "gradient-free". It is said that no gradient wrt the objective function is used, yet internally the optimizer computes gradients to perform back propagation. It needs to be described to which loss (objective) function these gradients are computed.

Response:

We agree with the reviewer that “gradient-free” tends to cause confusion since it seems to imply that there is no gradient in the method at all, while what this term really means is that the *gradient of objective function* is not used. Following the reviewer’s question, we have removed all “gradient-free” adjectives when referring to our method to avoid confusion. In the cases that we compare with other authors’ methods where other authors call their methods “gradient-free”, we have changed the term when referring to their methods as “non-gradient”. Although both terms “gradient-free” and “non-gradient” have been used interchangeably by different authors, we feel the term “non-gradient” is better since it is softer and is a direct contrast to term “gradient-based”. The term “non-gradient” implies that the optimization is not based on the gradient of objective function, and does not preclude the use of gradients in other parts of the optimization process. Therefore, we use the term “non-gradient” when referring to those methods that some other authors referred to as “gradient-free”.

As for the meaning of “non-gradient” (previously “gradient-free”), here is a detailed explanation. Assume we optimize the objective function $F(\rho)$. “Gradient-free” (or “non-gradient”) means that we do not use the information of $\nabla_{\rho}F(\rho)$. In our algorithm, we used a DNN $f_{\theta}(\rho)$ (parameterized by neuron weights θ) to approximate $F(\rho)$. We constructed a loss function $L(\theta) = (f_{\theta}(\rho) - F(\rho))^2$ to measure the distance between $f_{\theta}(\rho)$ and $F(\rho)$. To train the DNN, we need to calculate $\nabla_{\theta}L(\theta) = 2(f_{\theta}(\rho) - F(\rho))\nabla_{\theta}f_{\theta}(\rho)$ in order to minimize $L(\theta)$. As shown in the expression of $\nabla_{\theta}L(\theta)$, the gradient is calculated inside the DNN w.r.t. neuron weights, and the objective function $F(\rho)$ is only calculated at point ρ without gradient. The reason we quote “non-gradient” is that it has some benefits such as allowing $F(\rho)$ to be non-differentiable (e.g., discrete). Calculating the gradient internal of DNN does not void this property, and does not conflict with the definition of “non-gradient” since this term only limits the gradient of the objective function $\nabla_{\rho}F(\rho)$ (not $\nabla_{\theta}f_{\theta}(\rho)$).

Below is a summary of our changes regarding the term of “gradient-free”

- (1) Deleted all “gradient-free” adjectives when referring to our method. We only present this non-gradient property by a detailed description to avoid confusion: (Line 76) “**In contrast to gradient-based methods, this algorithm does not rely on gradient information of objective functions of the topology optimization problems. This property allows it to be applied to binary and discrete design variables in addition to continuous ones.**”

- (2) Replaced “gradient-free” by “non-gradient” when referring to the baseline methods by other authors. Although these two terms have the same definition and have been used interchangeably by different authors, “non-gradient” is softer and better. We appreciate that the reviewer has pointed to the confusion that the term “gradient-free” has caused.
- (3) Slightly expanded the general definition of “non-gradient”. In Line 29, we wrote that “**Non-gradient optimizers, also known as gradient-free or derivative-free methods, do not use the gradient or derivative of the objective function and have been attempted by several researchers...**”

2. *The additional examples need further explanations. It seems that the same truss structure is repeated to build the antenna. What exactly is optimized for in this example? Also for the copper-wax example it is not clear to me what the objective function is. Is there a volume constraint? Is compliance what you aim to optimize for?*

Response:

We thank the reviewer’s questions which help us to make the description clearer.

In the truss optimization problems, the same connection is repeated to build the antenna tower, but the diameter/size of the bars may be different (although they look the same in the figures). The objective is the weight of the truss, subject to stress and displacement constraints. The objective function is written as

$$\min_{\rho \in \{a_1, a_2, \dots, a_{16}\}^N} \widetilde{W}(\rho) = W(\rho)/W(\rho_{max}) = \sum_{i=1}^N \rho_i L_i \gamma_i / W(\rho_{max}), \quad (14)$$

where ρ_i , L_i and γ_i are the cross-sectional area, length, and unit weight of the i -th bar, respectively. ρ_{max} uses the largest cross-sectional area for all bars. Each bar is only allowed to choose from 16 discrete cross-sectional area values a_1, a_2, \dots, a_{16} , to represent standardized components in engineering applications. Constraints are needed in this problem, otherwise we would get a trivial solution where all bars reach the minimum. The stress constraint is applied to all bars:

$$|\sigma_i| \leq \sigma_0, \quad i = 1, 2, \dots, N. \quad (15)$$

The displacement constraint is applied to connections in any (x, y or z) direction

$$\|\Delta \mathbf{x}_i\|_\infty \leq \delta_0, \quad i = 1, 2, \dots, N_c, \quad (16)$$

where N_c is the number of connections.

To make the objective and constraints clearer for the truss problems, we add a sentence to explain: (Line 319) “**We set the goal to optimize the size of each bar (the bars can all have different sizes) to minimize total dimensionless weight...**” We summarized the problem setup by (Line 329) “**Now we have an optimization problem with objective Eq. (14) subject to stress constraint Eq. (15) and displacement constraint Eq. (16).**” Also, we briefed the setup in figure caption: (Fig. 7 caption) “**b, Illustration of the problem setup: minimizing total weight through changing the size of each bar, subject to stress and displacement constraints.**”

In the heat transfer enhancement problem (copper-wax example), the objective function is the charging time, i.e., the time to charge the system with a given amount of heat Q_0 :

$$\min_{\rho \in [0,1]^N} \tilde{t}(\rho) = t(\rho)/t(\rho_0), \quad (9)$$

$$\int_0^{t(\rho)} q(\rho, t) dt = Q_0, \quad (10)$$

where $\rho_0 = (0,0,\dots,0)$ means no copper inside the design domain, $q(\rho, t)$ denotes the inwards heat flux across the boundary.

We do have a volume constraint

$$\frac{\int_{\Omega} \rho(\mathbf{x}) d\mathbf{x}}{\int_{\Omega} d\mathbf{x}} = 0.2. \quad (13)$$

Additionally, we have a constraint on the heat flux across the boundary

$$q(\rho, t) \leq q_0, \quad (11)$$

and a constraint on maximum temperature

$$T(\rho, q, \mathbf{x}, t) \leq T_0, \quad (12)$$

where q_0 and T_0 are preset constants. Physically, Eqs. (11) and (12) mean that the system is charged at heat flux q_0 until the boundary temperature reaches T_0 or the total heat flow reaches Q_0 (whichever first), and if it is the former case, the heat flux is reduced to maintain the boundary temperature at T_0 until the heat flow requirement is satisfied.

For the heat problem, we revised the manuscript to help readers quickly grasp the objective and constraints. In Line 280, we wrote “Our goal is to find the optimal ρ to minimize the time to charge the system with a given amount of heat ...”. In Line 294, we added “To solve the problem with objective Eq. (9) and constraints in Eqs. (11)-(13)...” Besides, we added a brief description in Fig. 6 caption “**b, Problem setup: minimizing the time to charge the system with a given amount of heat, subject to heat flux, temperature and volume constraints.**” To explicitly show the volume constraint, we added (Line 286) “and given copper usage, i.e, the volume constraint of copper,”

3. I was irritated about the statement "We agree with the reviewer that the network has to be retrained for every topology optimization" What does this mean? Please comment on the implications of this in real-world application where usually new designs want to be developed and tested.

Response:

The sentence was in response to Question 1 by Reviewer #1 who asked about term “online” vs “the system has to be retrained for every topology optimization”.

We appreciate this question by Reviewer #2 so that we can highlight our method better. First, we would like to discuss the implications in real-world applications on a high level. It should be recognized that it is not feasible to train a DNN once and be able to solve all problems. Whenever a domain geometry or problem changes, retraining is needed. In other words, retraining cannot be avoided. In general (including our method and most other optimization methods), intermediate results (trained weights in our case) of optimization processes will be discarded after finding the optima. Training a DNN as part of our algorithm introduces a computation overhead, but it is negligible compared with other processes and beneficial considering the great reduction on overall computation. More details are shown below.

During the optimization process, we build a DNN $f_\theta(\rho)$ parameterized by neuron weights θ to approximate the objective function, i.e., $f_\theta(\rho) \approx F(\rho)$. Our algorithm leverages the minimum of the DNN $\hat{\rho} = \operatorname{argmin} f_\theta(\rho)$ to find the minimum of $F(\rho)$. After that, the DNN $f_\theta(\rho)$ can be discarded. If we have another topology optimization problem with objective function $G(\rho)$, we build another DNN $f_\eta(\rho)$ to approximate $G(\rho)$, i.e., $f_\eta(\rho) \approx G(\rho)$. The parameter set η is learned by retraining the DNN and thus has nothing to do with θ . Although in principle we could reuse θ to accelerate the training for η (for instance, by using θ as the initial point of η), we do not think it is worth the effort since the time to train a DNN is negligible. In Supplementary Table 1, we listed the computation time profile of the following three steps in a loop: (1) calculating ground truth $F(\rho)$, (2) training a DNN, and (3) finding the minimum of the DNN. In these steps, (1) takes most of the time, (3) becomes much shorter, and (2) is most negligible. Therefore, we do not think reusing the parameters from previous or similar problems is needed.

As for real-world applications where usually new designs want to be developed and tested, retraining does not cause any issue. In most cases, the results (e.g., weights of DNN, intermediate solutions) from old problems cannot be transferred to new problems. Our low overhead of training DNNs is particularly attractive since no transfer from old problems does not have a negative impact on the performance. For a given problem where we look for a new optimal design, we simply treat it as an independent problem without worrying about its relationship with other problems. In fact, almost all current optimization algorithms do not reuse intermediate results of other optimization processes.

Following the reviewer's comment, we added a sentence: (Line 360) “**Similar to other SMBO methods, overhead computation was introduced (by training DNNs and finding their optima), but it was almost negligible (see the time profile in Supplementary Table 1) which is attractive for real-world applications where new designs want to be developed and tested.**”

Reviewer #3:

It is appreciated that the authors spend a lot of effort in revising the manuscript.

The authors have addressed most of my concerns (1) discussion on related works; (2) real-world examples; (3) improvements on the organization and writing.

1. I agree with other reviewers that the phrase "gradient-free" used in the paper is quite confusing. Although the authors added a sentence for clarification, I think it can still be misleading.

Response:

We agree with the reviewer that “gradient-free” tends to cause confusion since it seems to imply that there is no gradient in the method at all, while what this term really means is that the *gradient of objective function* is not used. Following the reviewer’s question, we have removed all “gradient-free” adjectives when referring to our method to avoid confusion. In the cases that we compare with other authors’ methods where other authors call their methods “gradient-free”, we have changed the term when referring to their methods as “non-gradient”. Although both terms “gradient-free” and “non-gradient” have been used interchangeably by different authors, we feel the term “non-gradient” is better since it is softer and is a direct contrast to term “gradient-based”. The term “non-gradient” implies that the optimization is not based on the gradient of objective function, and does not preclude the use of gradients in other parts of the optimization process. Therefore, we use the term “non-gradient” when referring to those methods that some other authors referred to as “gradient-free”.

Below is a summary of our changes regarding the term of “gradient-free”

- (1) Deleted all “gradient-free” adjectives when referring to our method. We only present this non-gradient property by a detailed description to avoid confusion: (Line 76) **“In contrast to gradient-based methods, this algorithm does not rely on gradient information of objective functions of the topology optimization problems. This property allows it to be applied to binary and discrete design variables in addition to continuous ones.”**
- (2) Replaced “gradient-free” by “non-gradient” when referring to the baseline methods by other authors. Although these two terms have the same definition and have been used interchangeably by different authors, “non-gradient” is softer and better. We appreciate that the reviewer has pointed to the confusion that the term “gradient-free” has caused.
- (3) Slightly expanded the general definition of “non-gradient”. In Line 29, we wrote that **“Non-gradient optimizers, also known as gradient-free or derivative-free methods, do not use the gradient or derivative of the objective function and have been attempted by several researchers...”**

2. Overall, I still think that the contribution of the paper is limited. The results are incremental to the field, and the findings are not very exciting. So I believe the impact will be limited as well.

Response:

Following the reviewer's comment, we add more highlights on our contribution:

(Abstract, Line 10) “**Proof of convergence was given.**”

(Introduction, Line 86) “Our algorithm reduces the computational cost by at least two orders of magnitude compared with directly applying heuristic methods **including Generalized Simulated Annealing (GSA), Binary Bat Algorithm (BBA) and Bat Algorithm (BA).** It also outperforms an **offline version (where all training data are randomly generated), BO, CMA-ES, and a recent algorithm based on reinforcement learning³³.**”

Overall, we proposed an algorithm to solve topology optimization problems that current methods are unable or extremely expensive to address. We gave theoretical proof to show the algorithm is guaranteed to converge. Our algorithm reduced the computation by over two orders of magnitude compared with heuristic methods and outperformed all state-of-the-art model-based and deep-learning-based methods listed in the manuscript by a large margin. The algorithm can also be used in other fields for high-dimension optimization. We believe our method can help to solve the problems which have not been solved before and inspire the community to explore further on more advanced algorithms.

REVIEWERS' COMMENTS

Reviewer #2 (Remarks to the Author):

I think the manuscript has now reached a level where it can be published. My concerns have been discussed, some of them remain, some others have been waived.